# VPGTrans: Transfer Visual Prompt Generator across LLMs

Ao Zhang [1]    Hao Fei [1*]    Yuan Yao [2*]    Wei Ji [1]    Li Li [1]    Zhiyuan Liu [2]    Tat-Seng Chua [1]

[1] NExT++ Lab, School of Computing, National University of Singapore
[2] Department of Computer Science and Technology, Tsinghua University
zhanga6@outlook.com    haofei37@nus.edu.sg    yaoyuanthu@163.com

Project: https://vpgtrans.github.io

## Abstract

Since developing a new multimodal LLM (MLLM) by pre-training on a tremendous amount of image-text pairs from scratch is exceedingly resource-consuming, connecting an existing LLM with a comparatively lightweight visual prompt generator (VPG) becomes a feasible paradigm. However, further tuning the VPG component of the MLLM still incurs significant computational costs, such as thousands of GPU hours and millions of training data points. An alternative solution is to transfer an existing VPG from one MLLM to the target MLLM. In this work, we investigate VPG transferability across LLMs for the first time, aiming to reduce the cost of VPG transfer. Specifically, we explore VPG transfer across different LLM sizes (*e.g.*, small-to-large) and types. We identify key factors to maximize the transfer efficiency, based on which we develop a simple yet highly effective two-stage transfer framework, called **VPGTrans**. Notably, it enables VPG transfer from BLIP-2 $OPT_{2.7B}$ to BLIP-2 $OPT_{6.7B}$ with less than 10% of GPU hours using only 10.7% of the training data compared to training a VPG for $OPT_{6.7B}$ from scratch. Furthermore, we provide a series of intriguing findings and discuss potential explanations behind them. Finally, we showcase the practical value of our VPGTrans approach, by customizing two novel MLLMs, including VL-LLaMA and VL-Vicuna, with the recently released LLaMA and Vicuna LLMs.

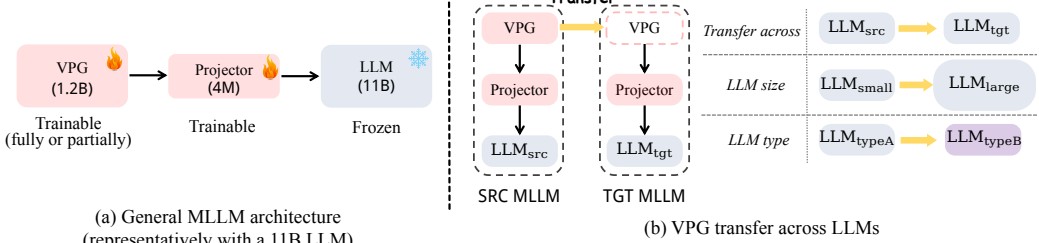

(a) General MLLM architecture
(representatively with a 11B LLM)

(b) VPG transfer across LLMs

Figure 1: (a) The general architecture of MLLMs, *e.g.*, BLIP-2 [30] and PaLM-E [16], including a visual prompt generator (VPG), a linear projector and a backbone LLM. Typically, to tune the MLLM, only the VPG and the projector are updated, while the LLM is kept frozen. (b) This work investigates the VPG transferability across LLMs, including different LLM sizes and LLM types.

## 1 Introduction

**Background.**    Recent years have witnessed a great rise in large-scale language models (LLMs) in ushering the human-like artificial intelligence. Text-based LLMs [42, 7, 44] are further enhanced

---

*Corresponding Author: Hao Fei, Yuan Yao

37th Conference on Neural Information Processing Systems (NeurIPS 2023).

by associating with other modalities such as vision, leading to the multimodal LLMs (MLLMs), such as BLIP-2 [30], Flamingo [2], GPT-4 [8] for multimodal dialog system, and PaLM-E [16] for embodied AI system. To construct a MLLM, a visual prompt generator (VPG) module (*cf.* Fig. 1(a)) that produces soft prompts for the input images/videos is added[2] for bridging the gap between vision and language modalities. Currently, such architecture has been frequently adopted by many popular MLLMs [30, 27]. For example, BLIP-2 pre-trains a CLIP-ViT [43] combined with a Q-Former as VPG. To obtain the final MLLM, the VPG needs to be tuned. Ideally, the vast LLM backbone can remain untouched, leaving only the relatively lightweight VPG module to be fully or partially updated.[3]

**Motivation.** However, building a MLLM is inevitably computation-expensive, due to the huge overhead brought by the LLM. For example, training a BLIP-2 FlanT5$_{\text{XXL}}$ needs over 600 A100-GPU hours on over 100 million image-text pairs. Hopefully, transferring a pre-trained VPG (which is the main body of trainable parts) from an existing MLLM to a novel LLM instead of training from scratch,[4] offers a promising solution. Intuitively, all the MLLMs literally can share the same VPG infrastructure and utility,[5] which makes the VPG transfer theoretically feasible. In this work, we thus investigate the potential of transferring VPG across LLMs.

**Proposal.** Specifically, this paper examines the transferability of VPG across LLMs: 1) with different sizes (the same type), *i.e.*, *transfer across LLM sizes*, and 2) across different LLM types, *i.e.*, *transfer across LLM type*, as illustrated in Fig. 1(b).

- [***Transfer across LLM Sizes*** (**TaS**)]. It has been a typical practice for LLM-related research [8] to validate the training strategy and the hyperparameter on smaller models (*e.g.*, OPT$_{2.7B}$) and then scale up to larger ones (*e.g.*, OPT$_{6.7B}$). It is thus worth exploring whether a VPG trained on a smaller LLM can be transferred to a larger LLM, resulting in reduced computational costs & data, and maintaining comparable performance.
- [***Transfer across LLM Types*** (**TaT**)]. With a well-tuned VPG for a type of LLM, it is interesting to see if VPG can be transferred to other types of LLMs even with different architectures (*e.g.*, decoder *v.s.* encoder-decoder). If the transfer can be achieved, how to make it more efficient?

We conduct a series of exploratory analyses (*cf.* §3.1) to identify the key factors for transfer efficiency. Based on our empirical study, we design a two-stage transfer learning framework (*cf.* §3.2), namely **VPGTrans**, that includes a projector warm-up (stage-1) and vanilla fine-tuning (stage-2). For stage-1, we find that warming up the projector before VPG tuning can effectively reduce the training step for adapting a pre-trained VPG to a new LLM, and avoid the potential performance drop in the adaptation. To achieve an efficient warm-up, the projector will be well-initialized and then trained with an extremely large learning rate ($5 \times lr$). For stage-2, there is a vanilla fine-tuning of both the VPG and projector. Despite its simplicity, VPGTrans is able to significantly speed up the VPG-transfer process without harming the performance.

**Results and Findings.** Via extensive experiments on the transfer across LLM sizes and types (*cf.* §4 & §5), we gain the following key observations:

- VPGTrans helps to avoid the performance drop caused by directly inheriting the VPG and achieves at most 10 times acceleration for the small-to-large transfer across LLMs in the same type.
- VPGTrans can also achieve comparable or better performance than training from scratch and achieve at most 5 times acceleration for the transfers between different model types.
- Notably, our VPGTrans helps to achieve a **BLIP-2 ViT-G OPT$_{2.7B \rightarrow 6.7B}$** transfer with **less than 10% of GPU hours** and **10.7% of training data** required for the original model training.
- Furthermore, our framework can even outperform the original BLIP-2 OPT$_{6.7B}$ on most of the evaluated datasets, with a **+2.9** improvement on VQAv2 and a **+3.4** improvement on OKVQA.

Our investigation further reveals some intriguing findings, for which we provide possible explanations:

---

[2]Also including a linear projector for dimension matching.

[3]Note that Flamingo also inserts some tunable parameters into the LLM part, but recent works [30, 16] found that freezing LLM can be more efficient.

[4]It is not a rigorous expression, because the VPG is typically a pre-trained model, like CLIP [43]. We use it for simplicity in this paper.

[5]while the projector can not be shared due to the dimension mismatch.

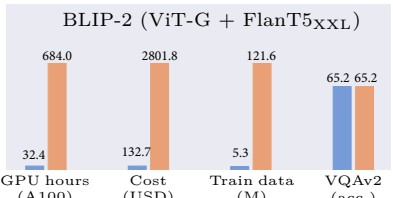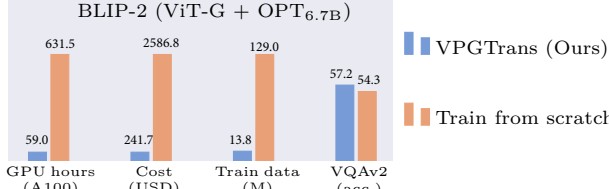

Figure 2: Comparing the cost between *training VPG from scratch* vs. *transferring VPG via our VPGTrans strategy*. Note the LLM via VPGTrans is FlanT5$_{XL \to XXL}$ and OPT$_{2.7B \to 6.7B}$, respectively.

- When conducting TaS from LLM$_{src}$ to LLM$_{tgt}$, the size of LLM$_{src}$ is not the larger the better. The transfer sometimes even follows a counterintuitive principle of "*the smaller the LLM$_{src}$ size, the more speed-up and better performance*" (*cf.* §4.2).
- When conducting TaT, efficient VPG transfer can not be achieved between two small LLMs with our VPGTrans, due to the large gap between small LLMs' embedding space (*cf.* §5.2).

**Contributions.** In this study, we show for the first time that effective VPG transfer across LLMs can be achieved under most conditions, suggesting that it is possible to build a new MLLM with considerably lower computational cost, as seen in Fig. 2. To summarize, we make the following key contributions:

- *Effective approach*. We investigate the key factors for VPG-transfer efficiency and propose a two-stage transfer framework VPGTrans. The approach helps to achieve a highly-efficient VPG transfer across LLMs with less training data and even task improvements.
- *Intriguing findings*. By exploring the VPG transfer across LLMs, we reveal several intriguing findings and provide potential explanations that will shed light on further research.
- *Open source*. We showcase how to customize a novel GPT-4-like MLLM with our VPGTrans (*cf.* §6), and release two multimodal-version MLLMs: VL-LLaMA and VL-Vicuna. All codes and models is released at `https://github.com/VPGTrans/VPGTrans`.

## 2 Preliminary

This section will outlines the existing prevailing MLLMs, and elaborates on the settings of the exploratory analyses of these MLLMs.

### 2.1 MLLM

**Architecture.** As illustrated in Fig. 1(a), current MLLMs mostly adopt a common architecture, including a visual prompt generator (VPG), a projector, and a backbone LLM. Typically, VPG takes images/videos as inputs, and encodes the visual input into a fixed length of soft prompts. Then, a linear projector is employed to align the soft prompt's dimension to LLM's word embedding dimension. Finally, the LLM will generate sentences based on the information from the soft prompt. We list some of the recent representative MLLMs in Table 1.

Table 1: MLLMs architectures and pre-training paradigm. †: it is a GPT-2-like LLM with relative position embeddings.

| MLLMs | VPG | VPG Trainable | LLM | LLM Trainable |
|---|---|---|---|---|
| KOSMOS-1 [19] | CLIP [43] | All | Rand. Init. LM | All |
| Frozen [54] | NF-ResNet-50 [6] | NF-ResNet-50 | GPT-2-like† [42] | No |
| Flamingo [2] | NFNet-F6 [6]+Resampler [21] | Resampler | Chinchilla [18] | Xattn-Dense |
| PaLM-E [16] | ViT [15] / OSRT [49] | All | PaLM [11] | No |
| BLIP-2 [30] | EVA-CLIP [52] + Q-Former [30] | Q-Former | OPT [60] / Flan-T5 [12] | No |

**Training Paradigm.** Given a MLLM, typically the VPG and linear projector will be trained, fully or partially. For example, PaLM-E updates all of the parameters of VPG in the pre-training stage, while BLIP-2 and Flamingo freeze the ViTs and tune their Q-Former and Resampler, respectively. As the main part of the whole architecture, the LLM is usually frozen during the training or tuned only a small portion (*e.g.*, 10B for Flamingo-80B). KOSMOS-1 is an exception, which does not use a pre-trained LLM but trains the LLM from scratch. Such a training paradigm typically results in much longer training time and data (both multimodal and pure text corpus). Recent works [30, 16] show that

adopting an existing LLM and freezing all of its parameters can also achieve excellent performance with significantly reduced computational cost, which leads to the trend of adapting frozen pre-trained LLM. For example, BLIP-2 FlanT5$_{XXL}$ (12.1B) can achieve better zero-shot VQAv2 performance (65.0% in Acc.) compared with KOSMOS-1 (51.0% in Acc.) and Flamingo-80B (56.3% in Acc.). Thus, in this paper, we mainly focus on VPG transfer across frozen LLMs.

## 2.2 Experiment Settings

**Architecture.** We adopt BLIP-2's architecture and training paradigm. In our exploration experiments, we consider using the VPG that consists of a CLIP ViT-L/14 [43], and a Q-Former that has already undergone a BLIP-like pre-training (the 1st stage pre-training in BLIP-2's paper [30]).

**Training Data.** For all of the exploration experiments, we adopt human-annotated COCO caption dataset [34] and web image-text pairs SBU dataset [40], which results in 1.4 million image-text pairs.

**Transfer Direction.** For the small-to-large model transfer among the same type of LLMs, we investigate: 1) OPT [60] (decoder-only) series including 125M, 350M, 1.3B, and 2.7B, and 2) FlanT5 [12] (encoder-decoder) ranging *base, large*, and *XL*. For the transfer across different types of LLMs, we consider the ones of OPT and FlanT5 with similar sizes.

**Evaluation.** To evaluate the performance of MLLMs, we choose two caption datasets: (1) COCO caption [34] (2) NoCaps [1], and three VQA datasets: (3) VQAv2 [4] (4) GQA [20] (5) OKVQA [37]. We make evaluations after the pre-training without task-specific fine-tuning and report the CIDEr [55] for all caption tasks and accuracy for all VQA tasks.

**Implementation Details.** We follow the same implementation details of BLIP-2, via the open code.[6] Concretely, we use FP16 and BFloat16 for OPT and FlanT5 respectively in the model training. For the learning rate, we first conduct a linear warm-up from 1e-6 to 1e-4, and then use a cosine learning rate schedule with the minimal $lr$=1e-5 for 10 epochs. Due to the limited data amount, we slightly decrease the batch size, which we find beneficial for the final performance. Specifically, we set the batch size of 1,728 and 1,152 for OPT and FlanT5-based models, respectively.

## 3 Maximizing the Transfer Efficiency with a Two-stage Transfer Strategy

In this section, we first identify the key factors for maximizing transfer efficiency, based on which we then motivate our solution for better transfer.

### 3.1 Exploratory Analysis: Identifying Key Factors for VPG Transfer

Via selected experiments of small-to-large transfer among OPT models, we can obtain the following key observations. More systematical comparisons are conducted in the later section (*cf.* §4).

• **Inheriting the trained VPG can accelerate training.** To demonstrate this, we compare the convergence rates of VPG training on OPT$_{350M}$ from scratch, and inheriting VPG trained on OPT$_{125M}$. The patterns are shown in Fig. 3. Overall, we find that inheriting VPG trained on OPT$_{125M}$ accelerates convergence, particularly for two caption tasks. However, for datasets that require fine-grained visual perception such as VQAv2 and GQA, directly conduct continue training with an inherited VPG will harm the performance. We hypothesize that tuning VPG with a randomly initialized projector will compromise the existing fine-grained visual perception ability of VPG. The possible reason can be that, the VPG is typically a pre-trained model with powerful visual perception ability, and thus updating based on the gradient passed through a random projector will mislead the VPG at the initial steps [2, 33, 23].

• **Warming up the linear projector can prevent performance drop and expedite VPG training.** To verify this, we first conduct a warm-up training of the linear projector for 3 epochs, during which both VPG and LLM are frozen. Subsequently, we jointly train VPG and the projector and plot the performance curve in Fig. 4 (**the warm-up process is not included in this figure**). The results show that the performance drop observed in Fig.3 can be avoided in Fig. 4. Additionally, we observe that the warm-up training leads to fewer training steps required for VPG and projector joint training. However, we must emphasize that warming up is a costly step. In the case of a large LLM, such as 6.7B, the trainable parameters of BLIP-2's VPG will account for less than 10% of the total parameters, where freezing VPG can only lead to a reduction of 5.4% of A100 hours (36.9 out of 684.0 A100

---

[6]https://github.com/salesforce/lavis

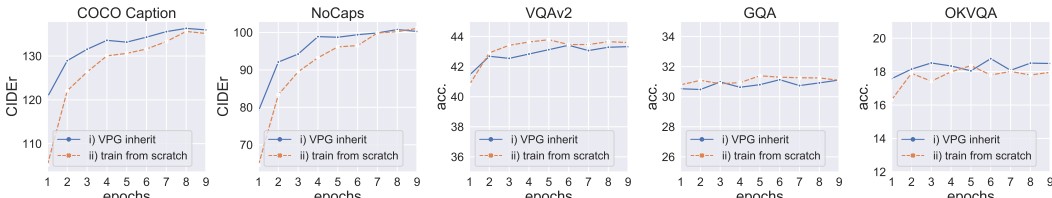

Figure 3: Comparisons between i) inheriting VPG from $OPT_{125M}$ and training it with randomly initialized projector for $OPT_{350M}$ and ii) training VPG and randomly initialized projector for $OPT_{350M}$ from scratch.

hours). We will elaborate on how to accelerate the linear projector warm-up in our later discussion (*cf.* 3.1).

● **Initializing LLM$_{tgt}$'s projector with the help of the word converter can accelerate the linear projector warm-up.** In fact, the VPG and projector trained on LLM$_{src}$ have already learned how to map the visual content to LLM$_{src}$'s understandable soft prompt [39]. If we can convert the LLM$_{src}$'s soft prompt to LLM$_{tgt}$'s soft prompt, we can directly get a VPG suitable for LLM$_{tgt}$. One natural idea is to leverage the word embeddings of both models as a proxy for the soft prompt [25]. The intuition behind the scene is that, the soft prompt works in the same format as normal words.

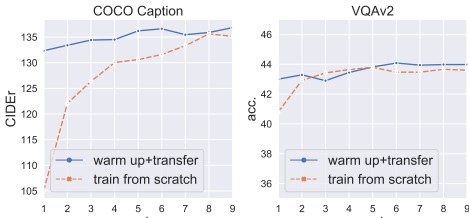

Figure 4: First warming-up then transferring can avoid performance drop on VQAv2 and accelerate convergence for COCO caption.

To validate our hypothesis, we conduct an experiment on the transfer from $OPT_{125M}$ to $OPT_{1.3B}$. After training a linear word embedding converter (*cf.* §3.2(b)), we initialize the projector for $OPT_{1.3B}$ with the merged linear operation of the projector for $OPT_{125M}$ and converter. As shown in Table 2, we observe that the initialization can reduce the 3 epochs' warm-up to 2 epochs.

● **Linear projector warm-up enables faster convergence with an extremely large learning rate.** To determine the most efficient transfer practice, we experiment with training the projector using different learning rates. Surprisingly, we find that the linear projector enables fast and stable convergence with an extremely large learning rate. Specifically, by setting the learning rate to 5 times of the original value, the COCO caption's CIDEr score can reach 133.1 with **1 epoch** training, which is **higher than the 3 epochs** results of *w/o init.* as shown in Table 2.

Table 2: Comparison between linear projector warm-up with/without word embedding initialization. The metric is COCO caption's CIDEr.

| Epoch | w/ init. | w/o init. |
|-------|----------|-----------|
| 1 | 130.2 | 126.1 |
| 2 | 132.7 | 131.6 |
| 3 | 133.4 | 132.8 |

### 3.2 A Two-stage VPG Transfer Framework

By connecting all the dots as discussed above in §3.1, we now design our two-stage VPGTrans framework for more efficient VPG transfer. As shown in Fig. 5, the stage-1 of VPGTrans performs projector warm-up and the stage-2 carries out a vanilla fine-tuning. Our results demonstrate that the VPGTrans is simple yet effective that can significantly speed up the transfer without compromising performance. Detailed results are given in the later sections (*cf.* §4 & 5).

▶ **Stage-1: Projector Warm-up.**

    *(a) Inherit VPG.* We first initialize the VPG for LLM$_{tgt}$ with the VPG trained on LLM$_{src}$.

    *(b) Projector Initialization.* Then, we initialize the projector for LLM$_{tgt}$ merged from the projector of LLM$_{src}$ and a linear word converter. Formally, we define the linear projector of LLM$_{src}$ as $f_s(x) = W_s x + b_s$, the linear projector for LLM$_{tgt}$ as $f_t(x) = W_t x + b_t$, and the word converter as $g_c(x) = W_c x + b_c$.

The word converter is a linear layer trained with text-only caption data to convert the LLM$_{src}$'s word embeddings to LLM$_{tgt}$'s word embeddings. We experiment with optimizing losses based on cosine similarity or Euclidean distance, and observe no significant difference between the two losses. Thus we simply use cosine similarity in our experiments. In cases where LLM$_{src}$ and LLM$_{tgt}$ use different

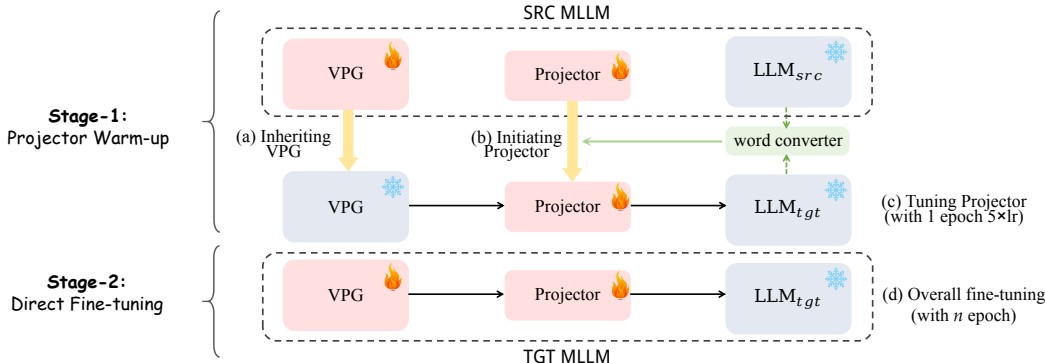

Figure 5: Our two-stage VPGTrans framework. **Stage-1** is to first (a) inherit the VPG of LLM$_{src}$ and (b) initialize the projector by merging the projector of LLM$_{src}$ and word converter. (c) Then the projector will be warmed up for 1 epoch with a large learning rate. **Stage-2** is to (d) conduct a vanilla fine-tuning for the VPG and projector for $n$ epochs.

tokenization methods, we optimize based on the overlapped tokens. Formally, for every given token $k$, we denote its word embeddings of LLM$_{src}$ and LLM$_{tgt}$ as $x_s$ and $x_t$. Then, we minimize the loss:

$$\mathcal{L} = 1 - sim(g_c(x_s), x_t). \tag{1}$$

Once we obtain the word converter $g_c(\cdot)$, we can easily merge it with the projector of LLM$_{src}$ as:

$$f_t(x) = f_s(g_c(x)) = W_s(W_c x + b_c) + b_s, \tag{2}$$

resulting in $f_t$'s weight and bias as $W_t = W_s W_c$ and $b_t = W_s b_c + b_s$.

*(c) Warm-up Training.* Then, we only train the projector in this stage with a frozen VPG and LLM. Specifically, we train the projector for 1 epoch with 5 times of the normal learning rate.

▶ **Stage-2: Vanilla Fine-tuning.**
  *(d) Vanilla Fine-tuning.* In the final step, we conduct a joint training of VPG and projector for $n$ epochs with a normal learning rate.

## 4 Exp-I: Transfer across Different Model Sizes

In this section, we conduct experiments to systematically illustrate the effectiveness of our VPGTrans and analyze the relationship between transfer efficiency and model size. For simplicity, we use **TaS** to represent the transfer across different model sizes.

### 4.1 Experimental Settings

In this part, we introduce baselines and transfer variants. For details about training data and implementation details, please refer to the experiment settings in the Preliminary (*cf.* 2.2).

**Baselines.**  We mainly compare our VPGTrans with *training from scratch* (TFS) and *VPG inheritance* (VPG Inherit), where we report their performance on the aforementioned 5 tasks without further task-specific fine-tuning. For our VPGTrans, the word converter training only requires updating a linear layer on tokenized text data and typically takes less than 10 minutes on 1 A100 GPU with less than 15G GPU memory. Meanwhile, freezing the VPG can lead to at least 14 A100 minutes speed-up per epoch. **Therefore, we consider the whole stage-1 training as the 1st epoch for simplicity**.

**Transfer Variants.**  We conducted experiments on transfer learning using 1) the OPT model across four different sizes: 125M, 350M, 1.3B, and 2.7B, and 2) the FlanT5 model across three sizes: *base, large*, and *XL*. However, we encountered significant instability during training with FlanT5$_{large}$. As a result, we mainly present the transfer results between FlanT5$_{base}$ and FlanT5$_{XL}$.

### 4.2 VPGTrans Enabling Faster Convergence without Performance Drop under TaS

First of all, as shown in Fig. 6, our VPGTrans can consistently accelerate the model convergence. For COCO caption and NoCaps that require more training steps to converge, our VPGTrans (green line) can be higher than the other two lines (blue and orange lines). To give a quantitative evaluation of the

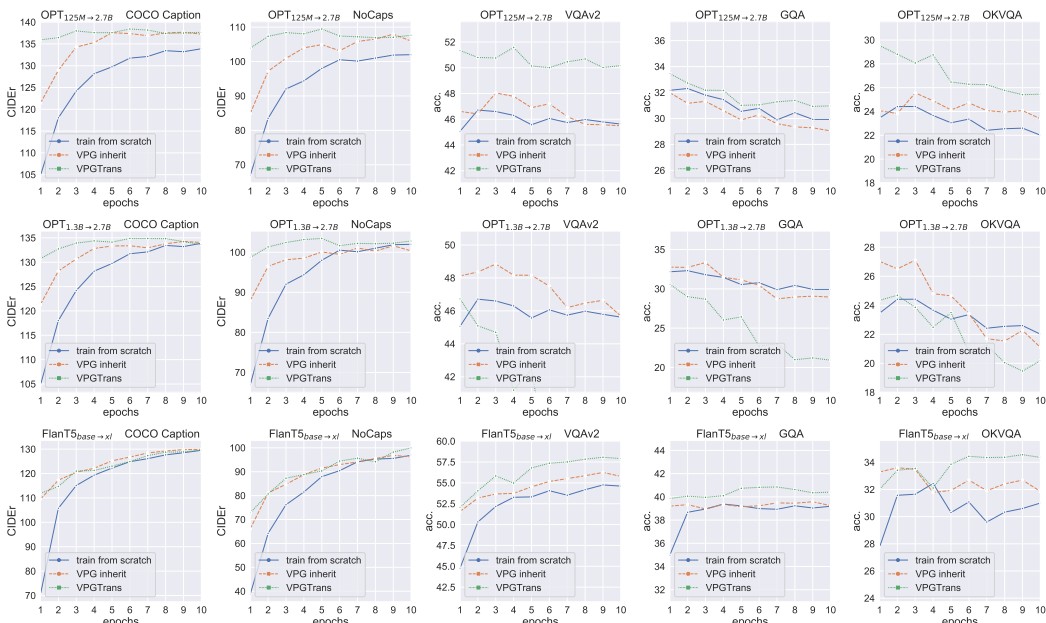

Figure 6: Comparison between different methods across 3 TaS variants on 5 tasks. Note that the model is directly evaluated after pre-training without further fine-tuning. Please refer to Appendix§C for other transfer variants.

Table 3: The speed-up rate of our VPGTrans compared with *training from scratch* (TFS). The symbol "-" means VPGTrans can not achieve better performance than TFS.

| Transfer | COCO Caption | NoCaps | VQAv2 | GQA | OKVQA |
|---|---|---|---|---|---|
| $OPT_{125M \to 350M}$ | 1.7 | 3.0 | 1.0 | 5.0 | 5.0 |
| $OPT_{125M \to 1.3B}$ | 9.0 | 10.0 | 9.0 | - | 2.0 |
| $OPT_{350M \to 1.3B}$ | 4.5 | 5.0 | 9.0 | 2.0 | 2.0 |
| $OPT_{125M \to 2.7B}$ | 10.0 | 10.0 | 2.0 | 2.0 | 3.0 |
| $OPT_{350M \to 2.7B}$ | 10.0 | 10.0 | 2.0 | - | 3.0 |
| $OPT_{1.3B \to 2.7B}$ | 3.3 | 3.3 | 2.0 | - | 1.5 |
| $FlanT5_{base \to XL}$ | 1.0 | 1.1 | 3.0 | 4.0 | 2.0 |
| $FlanT5_{XL} \to OPT_{2.7B}$ | 5.0 | 5.0 | 2.0 | 2.0 | 3.0 |
| $OPT_{2.7B} \to FlanT5_{XL}$ | 1.7 | 2.0 | - | 2.0 | - |

speed-up rate, we show the speed-up rate in Table 3. The speed-up rate is calculated by considering the number of epochs reduced to achieve the best TFS performance on a particular dataset. Formally, given a dataset $D$, TFS obtains the best performance $p$ on $D$ at epoch $e_{tfs}$, whereas VPGTrans first achieves a better performance than $p$ at epoch $e_{vt}$. The speed-up rate on $D$ is given by $\frac{e_{tfs}}{e_{vt}}$. According to Table 3, our VPGTrans can achieve at least 4 times speed-up on 40% of Transfer-Task variants. Furthermore, for the two caption datasets, which take a long time to converge, our VPGTrans $OPT_{125M \to 2.7B}$ delivers a 10 times speed-up.

Moreover, when compared with the *VPG inherit* in Fig. 6, our VPGTrans can achieve a higher speed-up rate on all of the variants on caption tasks, and achieve better performance on most variants except for $OPT_{1.3B \to 2.7B}$. We refer the readers to Appendix§C.3 for more comparisons.

We provide interesting findings with respect to the efficiency transfer by VPGTrans in the following.

• **The smaller size of LLM$_{src}$, the easier the transfer.** In our OPT based experiments, we notice an interesting phenomenon: when transferring to a given LLM$_{tgt}$, both the convergence rate and optimal performance are roughly inversely proportional to the size of LLM$_{src}$. For example, as shown in Table 3, the $OPT_{125M \to 2.7B}$ and $OPT_{350M \to 2.7B}$ have much higher speed-up rate than $OPT_{1.3B \to 2.7B}$ on all of the datasets. Meanwhile, as demonstrated in Fig. 6, the optimal performance of $OPT_{125M \to 2.7B}$ is better than $OPT_{1.3B \to 2.7B}$ on 3 VQA tasks.

Table 4: Comparison between models built with our VPGTrans and the original BLIP-2 ViT-G $OPT_{6.7B}$ and BLIP-2 ViT-G FlanT5$_{XXL}$.

| Models | VQAv2 val | GQA test-dev | OKVQA test | GPU hours | training data |
|---|---|---|---|---|---|
| BLIP-2 ViT-G OPT$_{6.7B}$ | 54.3 | 36.4 | 36.4 | 631.5 | 129M |
| BLIP-2 ViT-G OPT$_{2.7B \to 6.7B}$ (**ours**) | 57.2 | 36.2 | **39.8** | **59.0** | **13.8M** |
| VL-LLaMA$_{7B}$ (**ours**) | **58.1** | **37.5** | 37.4 | 67.1 | 13.8M |
| BLIP-2 ViT-G FlanT5$_{XXL}$ | 65.2 | 44.7 | **45.9** | 684.0 | 121.6M |
| BLIP-2 ViT-G FlanT5$_{XL \to XXL}$ (**ours**) | **65.2** | **45.0** | 45.0 | **32.4** | **5.3M** |

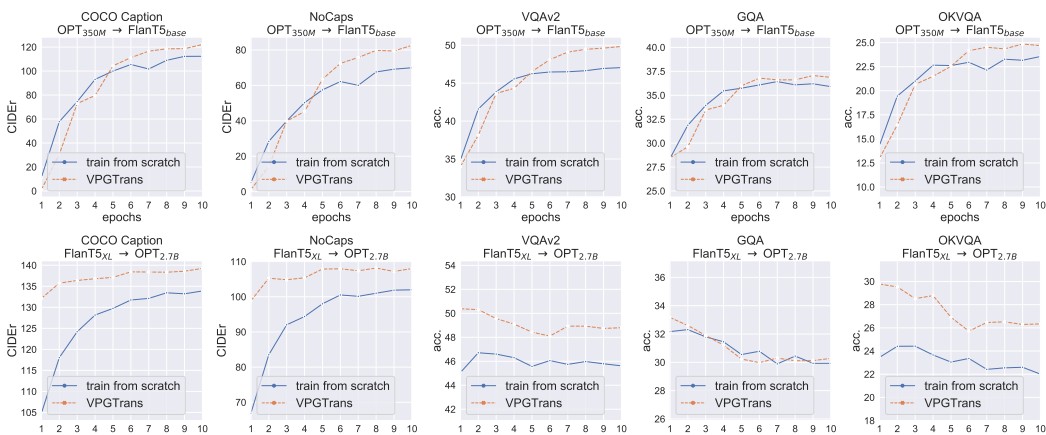

Figure 8: Comparison between different methods across 2 TaT variants on 5 tasks. Note that the model is directly evaluated after pre-training without further fine-tuning. Please refer to Appendix§D for other transfer variants.

We hypothesize that training VPG on larger OPT will have a worse influence on VPG's existing fine-grained perception ability, which might be caused by the enlarging embedding dimensions. To validate our hypothesis, we fix the VPG weight and only tune linear projectors to test VPGs trained on different $LLM_{src}$ through cross-size transfer. The SPICE [3] metric on COCO caption is used to evaluate the VPG's visual perception ability, where SPICE is specifically designed for visual concept perception in captions. As shown in Fig. 7, for each row, given the $LLM_{tar}$, the performance of VPG trained on smaller $LLM_{src}$ can outperform the larger ones in most conditions, which indicates a better visual perception ability of VPG trained on smaller $LLM_{src}$. Therefore, **adapting a VPG from a smaller OPT model which is less affected, is helpful to take fewer steps to reach the TFS's best performance and achieve even better performance.**

### 4.3 Scale-up Experiments

To validate the effectiveness of our VPGTrans on the real-world application level, we experiment on transferring from BLIP-2 ViT-G OPT$_{2.7B}$ to OPT$_{6.7B}$ and from BLIP-2 ViT-G FlanT5$_{XL}$ to FlanT5$_{XXL}$. Please refer to Appendix§C.4 for implementation details.

**Speed-up with non-degenerated performances.** As shown in Table 4, we can see that **(1) OPT$_{2.7B \to 6.7B}$:** our VPGTrans achieves a 10.7 times speed-up with only 10.7% training data, while the performance on VQAv2 and OKVQA have over 2 points improvement. **(2) FlanT5$_{XL \to XXL}$:** VPGTrans can achieve 21.1 times speed-up with less than 5% training data while achieving the same performance on VQAv2, higher performance on GQA and slightly lower performance on OKVQA. Note that continuing training the FlanT5$_{XL \to XXL}$ only shows improvement on VQAv2 and GQA. Thus, we do not show a checkpoint with more training steps.

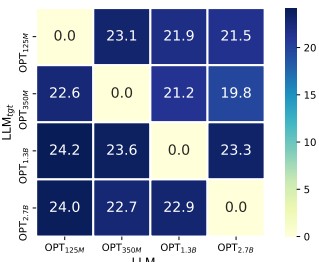

Figure 7: The confusion matrix. Only linear layers are trained for VPG evaluation. Models are tested on COCO caption with SPICE metric to compare the VPGs trained on different $LLM_{src}$.

Table 5: Comparison between our VL-Vicuna and SOTA MLLMs for multimodal conversation. The evaluation is done by Multimodality Chatbot Arena platform via user voting.

| Rank | Model | Elo Rating |
|---|---|---|
| 1 | LLaMA-Adapter v2 [17] | 1023.0 |
| 2 | LLaVA [35] | 1019.9 |
| 3 | **VL-Vicuna (ours)** | 1012.1 |
| 4 | MiniGPT-4 [62] | 1011.9 |
| 5 | InstructBLIP [13] | 999.5 |
| 6 | mPLUG-Owl [59] | 996.3 |
| 7 | Otter [26] | 981.5 |
| 8 | BLIP-2 [30] | 955.8 |

## 5  Exp-II: Transfer across Different Model Types

In this section, we further investigate the transfer across different model types. For simplicity, we mark this type of transfer as **TaT**.

### 5.1  Experimental Settings

In this part, we introduce baselines and transfer variants. For details about training data and implementation details, please refer to the experiment settings in the Preliminary (*cf.* 2.2).

**Baselines.**  We mainly compare our VPGTrans with *training from scratch* (TFS), and report the performance on the aforementioned 5 tasks. Other details (*cf.* 4.1) are totally the same with TaS experiments.

**Transfer Variants.**  We conducted experiments on transfer learning between 1) $OPT_{350M}$ and $FlanT5_{base}$, and 2) $OPT_{2.7B}$ and $FlanT5_{XL}$.

### 5.2  VPGTrans Enabling Faster Convergence only on Large LLMs under TaT

• **There is no speed-up of TaT between two small LLMs.**  A finding is that on TaT our VPGTrans does not show speed-up for small models, and even shows a degeneration of training speed in the initial several epochs. As shown in Fig. 8, when transferring from $OPT_{350M}$ to $FlanT5_{base}$, the convergence speed of VPGTrans is even slower than TFS in the initial several epochs.

• **Speed-up of VPGTrans happens in large LLMs.**  However, when moving to the large LLMs like $OPT_{2.7B}$ and $FlanT5_{XL}$, there is an obvious speed-up. As shown in Table 3, we can see at least 2 times speed-up when transferring from $FlanT5_{XL}$ to $OPT_{2.7B}$. We empirically find that **the soft prompts for larger LLM are more linear transferrable among different LLM types**. As shown in Fig. 8, when transferring between $FlanT5_{base}$ and $OPT_{350M}$, the VGPTrans' 1st epoch results on two caption datasets are limited, where only a linear operation can be trained. The result of $OPT_{350M} \rightarrow FlanT5_{base}$ on the COCO caption is even near to zero. By contrast, the result of $FlanT5_{XL} \rightarrow OPT_{2.7B}$ with our VPGTrans are obviously higher than TFS. We hypothesize that larger LLM typically learned more generalizable text embeddings and share more similarity among relative word distances, which enables an easier VPG transfer.

## 6  Customizing New MLLMs with Any LLMs

Above, we thoroughly certify the efficacy of our proposed VPGTrans approach for higher efficient transfer of VPG. In this section, we illustrate how to apply the VPGTrans framework for VPG transfer to customize new MLLMs with any LLMs.

**VL-LLaMA.**  By applying our VPGTrans, we can equip the recently released LLaMA [53] model with a VPG trained on BLIP-2 $OPT_{6.7B}$ to perceive the visual information. As shown in Table 4, we can see that our VL-LLaMA can outperform the original BLIP-2 $OPT_{6.7B}$ on all datasets.

**VL-Vicuna.**  An exciting application of our VPGTrans is to build a GPT-4 [8] style multimodal conversation chatbot. To achieve our goal, we employ Vicuna [10] as our base LLM. Similarly, we transfer the VPG from BLIP-2 $OPT_{6.7B}$, and add an extra instruction tuning using MiniGPT-4's self-instruct data [62]. We compare our model with MiniGPT-4 in Fig. 9. When compared to MiniGPT-4, our VL-Vicuna shows better visual perception ability. Please refer to Appendix§E for more cases. In

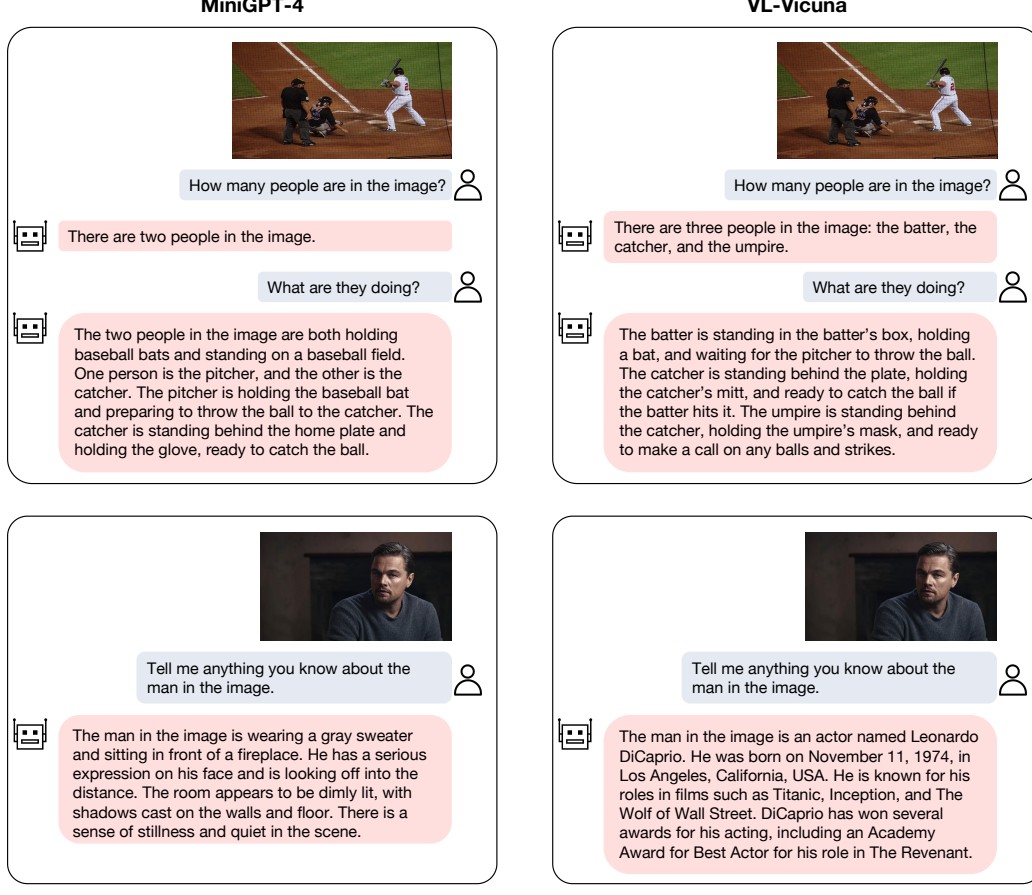

Figure 9: Comparison between MiniGPT-4 and our VL-Vicuna.

addition, we also report the ranking and ELO ratings for our VL-Vicuna compared with the other 7 SOTA multimodal chatbots in Table 5 (The evaluation is done by Multimodality Chatbot Arena platform[7]). The results show the effectiveness of our VL-Vicuna compared with existing SOTA models.

## 7 Conclusion

In this work, we conduct a comprehensive investigation to the problem of VPG transferability across LLMs. We first explore the key factors for maximizing the transfer efficiency under the VPG transfer across different LLM sizes and types. Based on the key findings, we propose a novel two-stage transfer framework, namely VPGTrans, which can help to achieve comparable or better performance while significantly reducing the training costs. Moreover, a list of important findings and possible reasons behind them are shown and discussed. Finally, we demonstrate the practical value of our VPGTrans, by customizing new MLLMs via VPG transfer from existing MLLMs.

## Acknowledgments

This research is supported by NExT++ Lab, Singapore Ministry of Education Academic Research Fund Tier 2 under MOE's official grant number T2EP20221-0023, and CCF-Baidu Open Fund.

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

# A    Related Work

## A.1    Vision and Language Models

Vision and language models (VLMs) [50, 29, 32, 31] aim at understanding visual and textual information with a single model. Previously, the VLM mainly employs a pre-trained object detector as its feature extractor and conducts unsupervised training on a huge amount of image-text pairs. For example, VL-Bert [50], ViL-Bert [36] and Uniter [9] adopt Faster-RCNN [47] to extract image information into object features and take advantage of the masked language model as their pre-training task. Later, due to the prevalence of vision transformer (ViT), the VLM paradigm is turned into end-to-end training with a ViT as the visual encoder. The representative works include ALBEF [28], BLIP [29], and BEIT-v3 [56], which show the state-of-the-arts supervised training performance on a wide range of downstream tasks.

Recently, LLMs have shown their remarkable capability as zero/few-shot learners [7] and a series of emergent abilities [57] like in-context learning [7], and chain-of-thoughts reasoning [58]. A new paradigm, i.e., MLLMs is created by associating the VLM or pure vision encoders with LLMs. As we illustrated before, the VLM or visual encoders are typically able to convert the input vision signals into LLM-understandable soft prompts, and thus we call them VPG. The MLLMs advance in inheriting the great potentials of the backbone LLMs, and thus are capable of achieving excellent zero/few-shot performances [2, 30] on downstream tasks or be equipped with visual planning ability [16]. However, connecting the VPG to the existing LLMs with further tuning is costly. Even the BLIP-2 [30], targeted at efficient training, will take over 600 A100 GPU hours on over 100M image-text pairs for its largest model. With this regard, our proposed VPGTrans can effectively reduce the cost of building new MLLMs with the help of existing ones.

## A.2    Prompt Transfer

In this paper, we investigate the VPG transfer, where the soft prompt is to represent the content of specific inputs like images and videos. In addition to the content prompt, the more explored soft prompt is the task prompt [24, 61, 22], where a sequence of soft prompts are tuned to assist the pre-trained models to achieve better performance on specific tasks. There have already been some works exploring the transferability of task prompts. For example, Su et al. [51] conducts a series of experiments to illustrate the transferability across tasks and models. Specifically, Su et al. [51] find that the transfer between similar tasks is beneficial for training speed-up and better performance. Lester et al. [25] proposes to recycle soft prompts across models with vocab-to-vocab transformations, or linear-combination transformations. Note that our word converter initialization is similar to the idea of vocab-to-vocab transformations. However, we do not observe a zero-shot transfer in our experiments like them, which indicates a potential difference between the content prompts and task prompts. Another way of soft prompt transfer [14] is to conduct prompt tuning on discrete prompts, and thus the discrete prompts can be directly shared across models. Different from these task prompts transfer works, our VPG transfer scenario actually suffers from fewer limitations. For example, the task soft prompts transfer suffers from the dimension change problem, where the main body of the trainable parameters should be processed. However, our VPG (the main trainable parameters) can naturally be shared among LLMs with different embedding dimensions and leave the dimension change problem to a simple projector with ignorable parameters.

# B    Extended Findings in Exploratory Analysis

In this section, we show extended findings of exploratory analysis (*cf.* §3.1).

• **1. Merely tuning the projector can not achieve the best performance.**    We want to clarify that merely tuning the projector is insufficient for achieving the best performance. Notably, as shown in Fig. 10, significant performance gaps are observed between the "*only linear*" (green curve) and "*train from scratch*" (orange curve) approaches for COCO caption and NoCaps. Therefore, if the goal is to build a multimodal conversation robot using carefully collected dialog data, training only the linear projector is insufficient to align with the provided data.

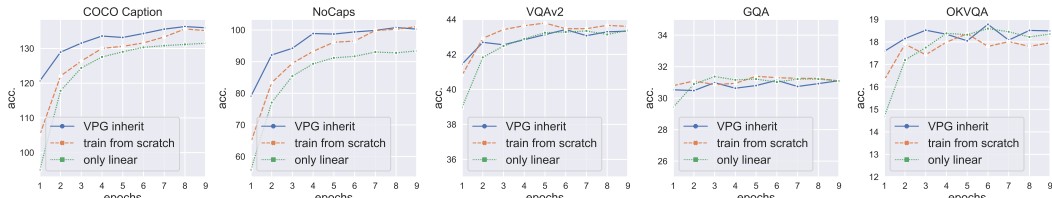

Figure 10: Comparisons between i) inheriting VPG from OPT$_{125M}$ and training it with randomly initialized projector for OPT$_{350M}$ and ii) training VPG and randomly initialized projector for OPT$_{350M}$ from scratch. iii) training only the projector.

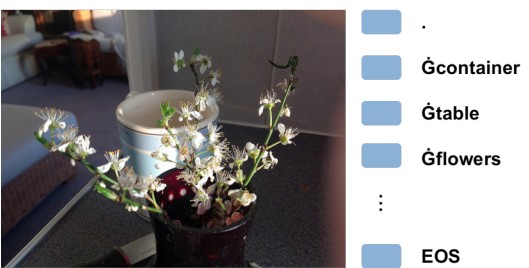

Figure 11: Interpreting generated soft prompts for OPT$_{125M}$ with nearest words.

• **2. Word embedding converter can not replace a trained linear projector.** As demonstrated in Fig. 11, we observe a common pattern: the last token of soft prompts is closest to EOS, while the middle tokens represent the image content. Such a phenomenon indicates a similarity between soft prompts and word embeddings. However, they are not identical. For instance, the norm of soft prompts is typically around 10 times the average norm of word embeddings. It is important to note that the linear projector initialization cannot replace the warm-up training. Using only the linear projector initialization even yields a random performance. We believe that a better understanding of how prompt works will further benefit the VPG's transfer learning.

• **3. The projector warm-up is robust to a larger learning rate, while VPG can not.** The first thing we want to clarify is that the 5 times normal learning rate will result in a training crash for VPG. Additionally, we find that although increasing the learning rate to 10 times in the projector warm-up does not yield any additional acceleration, the projector can converge without crashing during training.

## C  Extended TaS Experiments

In this section, we first illustrate extending findings of TaS experiments (*cf.* §4). Then, we introduce the implementation details of scale-up experiments. We also plot a more complete version of Fig. 6 in Fig. 13.

### C.1  VPGTrans Enabling Stable Training under TaS

As we illustrated before, FlanT5$_{large}$ training is extremely unstable. Even when the learning rate is adjusted to one-tenth of its original value, the model does not converge or shows a very slow convergence rate after 4 epochs. However, we find that by lowering the learning rate for stage-2 training, our VPGTrans can achieve stable training on FlanT5$_{large}$. We plot the performance curve of the COCO caption in Fig. 12.

### C.2  VPGTrans Enabling Training with Less Data under TaS

We empirically find that TaS can reduce the requirement for the amount of training data. By reducing the training data to only COCO, we find no obvious performance drop. However, we want to stress that **the retained data should be of high quality**, which means that if the same number of SBU data

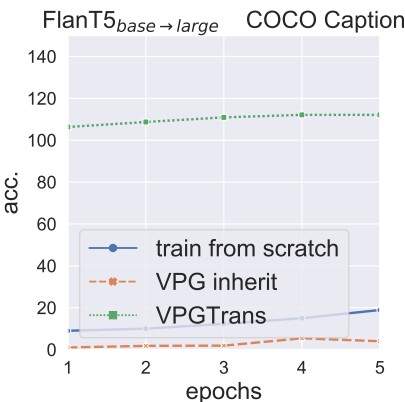

Figure 12: Comparison of training FlanT5$_{large}$ using different methods.

Table 6: The speed-up rate of our VPGTrans compared with *VPG inherit*. The symbol "-" means VPGTrans can not achieve better performance than *VPG inherit*.

| Transfer | COCO Caption | NoCaps | VQAv2 | GQA | OKVQA |
|---|---|---|---|---|---|
| OPT$_{125M \to 350M}$ | 1.1 | 2.7 | 10.0 | 6.0 | 6.0 |
| OPT$_{125M \to 1.3B}$ | - | 2.7 | - | - | - |
| OPT$_{350M \to 1.3B}$ | - | 1.7 | 1.0 | 2.0 | 1.0 |
| OPT$_{125M \to 2.7B}$ | 3.0 | 3.0 | 3.0 | 1.0 | 3.0 |
| OPT$_{350M \to 2.7B}$ | 3.3 | 4.5 | 1.0 | 1.0 | 1.0 |
| OPT$_{1.3B \to 2.7B}$ | 2.3 | 3.0 | - | - | - |
| FlanT5$_{base \to XL}$ | - | 1.0 | 1.8 | 9.0 | 0.4 |

is retained, the performance especially for captioning will drop. The conclusion can also be found in Table 4. We refer the readers to the next subsection for more details.

## C.3 Comparison between VPGTrans and VPG Inherit

As shown in Fig. 13, the green line (our VPGTrans) can be higher than the orange line (VPG inherit) for the majority of various conditions. Especially when considering the best performance of different tasks, our VPGTrans can achieve better performance than VPG inherit (non "-" in Table 6) for over 74% Transfer-Task variants. Moreover, among the variants that our VPGTrans can achieve better performance, our VPGTrans can also achieve a speed-up on 69.2% conditions.

## C.4 Scale-up Experiment Implementation Details

The scale-up experiment refers to the results in Table 4. We try to imitate BLIP-2's pre-training data composition. First of all, two human-annotated datasets **COCO** and **VG** are used. **SBU** is also used. Then, BLIP-2 uses BLIP to generate captions for the 115M web images and rank them with CLIP ViT-L/14. We also adopt similar synthetic data from **Laion-COCO**.[8] We report the concrete number of data we use in Table 4. For the stage-1 training, we keep the same as the previous validation experiments where COCO and SBU are used for warm-up with a 5 times the learning rate. Then, we use COCO, VG, and Laion-COCO for the stage-2 training. Note that we have tried to include Laion-COCO and VG for the stage-1 training, but found no obvious difference and thus use COCO and SBU for simplicity. For VL-Vicuna, to align with the conversation scenario, we further fine-tune our VL-Vicuna with MiniGPT-4's self-instruct data, which is 3,439 image-text pairs.

---

[8] https://laion.ai/blog/laion-coco

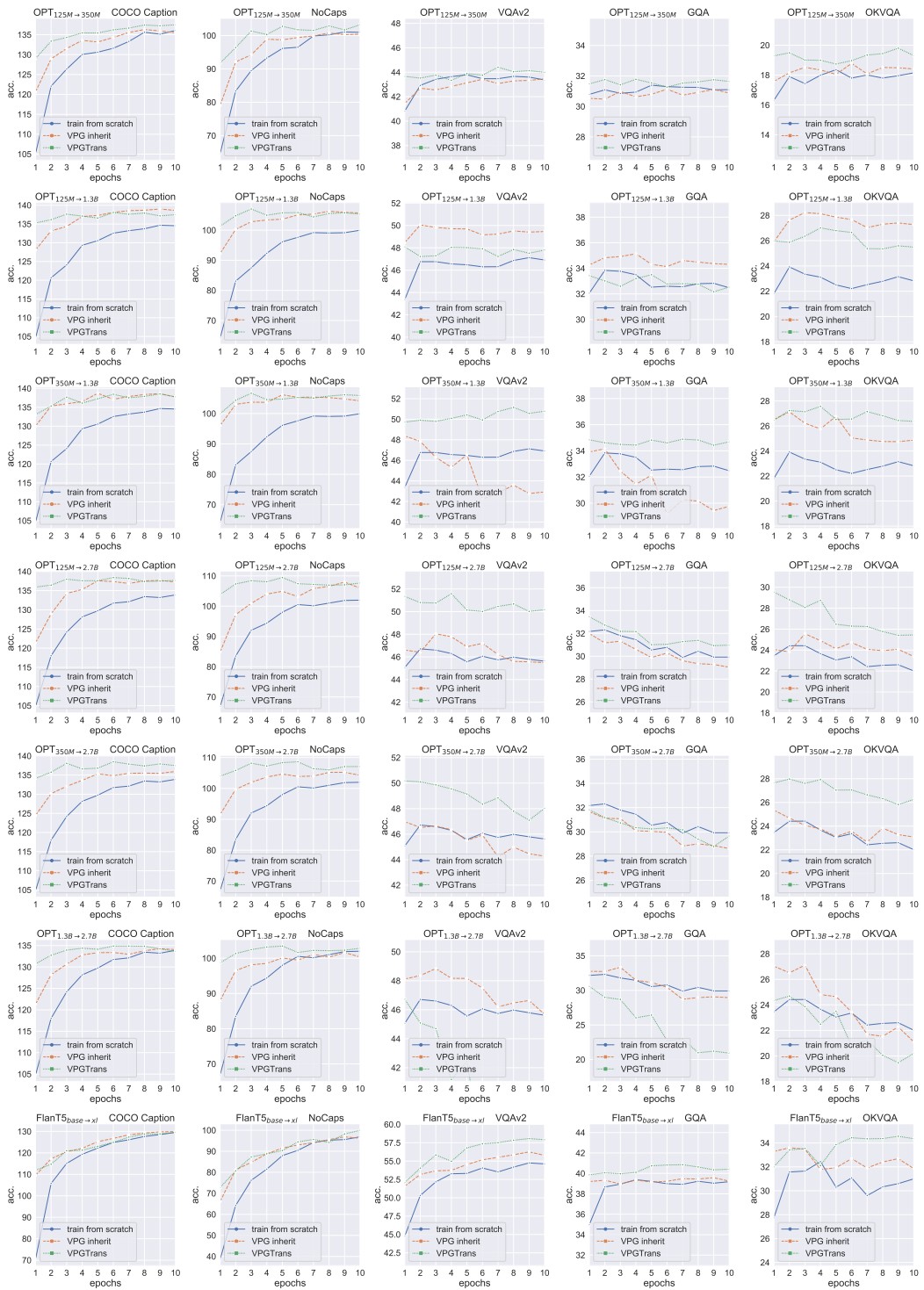

Figure 13: Comparison between different methods across 7 TaS variants on 5 tasks. Note that the model is directly evaluated after pre-training without further fine-tuning.

Table 7: Comparison between tuning only the projector (*i.e.* linear transfer) and the best results the VPGTrans achieve.

| Transfer | COCO Caption | | VQAv2 | |
|---|---|---|---|---|
| | linear | best | linear | best |
| $FlanT5_{base} \rightarrow OPT_{350M}$ | 110.8 | 136.5 | 40.4 | 44.2 |
| $FlanT5_{XL} \rightarrow OPT_{2.7B}$ | 132.1 | 139.3 | 50.4 | 50.3 |
| $OPT_{350M} \rightarrow FlanT5_{base}$ | 1.1 | 122.1 | 34.0 | 49.9 |
| $OPT_{2.7B} \rightarrow FlanT5_{XL}$ | 106.5 | 133.2 | 51.3 | 53.5 |

# D    Extended TaT Experiments

In this section, we mainly illustrate extending findings of TaT experiments (*cf.* §5). We plot a more complete version of Fig. 8 in Fig. 14.

### D.1    Linear Transfer Gap between Different LLM's Visual Prompts

As illustrated in Section 5.2, it is more difficult to transfer between two small LLMs with our VPGTrans due to the weaker linear transferability between two small LLMs' visual prompts. To better support our results, we compare the results of tuning only the projector (*i.e.* linear transfer) and the best results the VPGTrans can achieve. As shown in Table 7, we can see that when conducting transfers between two large models ($OPT_{350M}$ and $FlanT5_{base}$), the performance is typically far from the optimal results. However, when we transfer between $OPT_{2.7B}$ and $FlanT5_{XL}$, the linear performance is near to the optimal performance. There is a 25.7 points gap between $FlanT5_{base} \rightarrow OPT_{350M}$'s *linear* and *best* on COCO caption datasets but only 7.2 points gap between $FlanT5_{XL} \rightarrow OPT_{2.7B}$'s *linear* and *best*. If considering the transfer between the small to large LLMs under TaT, both the VPG's visual perception ability and transfer gap should be considered. We leave the systematical exploration for future works.

# E    Extended Results for MLLM Customization

We show more comparisons between VL-Vicuna and MiniGPT-4 in Fig. 15. We can see that our VL-Vicuna has better visual perception ability. For example, when MiniGPT-4 falsely recognizes the three people in the image as two in the first example, our VL-Vicuna can not only recognize the number of people but also tell their roles. Moreover, our VL-Vicuna can successfully link the vision content with external knowledge. In the third example, our VL-Vicuna can recognize Leonardo and link the content with his films like Titanic.

# F    Potential Impact and Limitations

Our VPGTrans is designed for building new MLLMs with lower computational cost, i.e., shorter training time and less training data. With an already pre-trained MLLM, VPGTrans enables fast VPG transfer to build either a larger MLLM or a MLLM with a different type of LLM. We hope VPGTrans can facilitate teams in LLM communicty to customize their MLLMs with reduced cost. There are also possible limitations of the current version of VPGTrans. The first one is that our VPGTrans should rely on some already-aligned VPGs. The second potential limitation is that the VPGTrans-built MLLMs still suffer from the common problems of content generation AI systems [45, 46, 48, 7]. For example, the VL-Vicuna may make up some sentences with falsely recognized visual facts, like what is shown in Fig. 16. It is worth exploring associating our VPGTrans with training safer models [5, 41, 38].

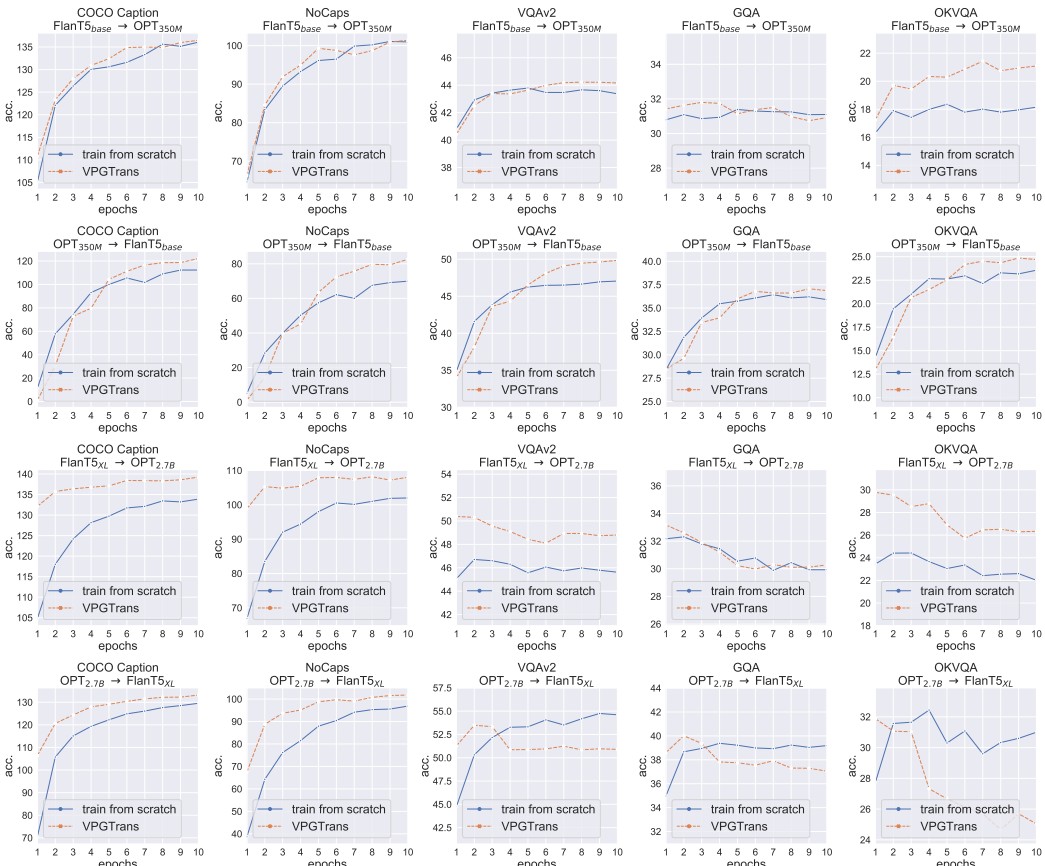

Figure 14: Comparison between different methods across 4 TaS variants on 5 tasks. Note that the model is directly evaluated after pre-training without further fine-tuning.

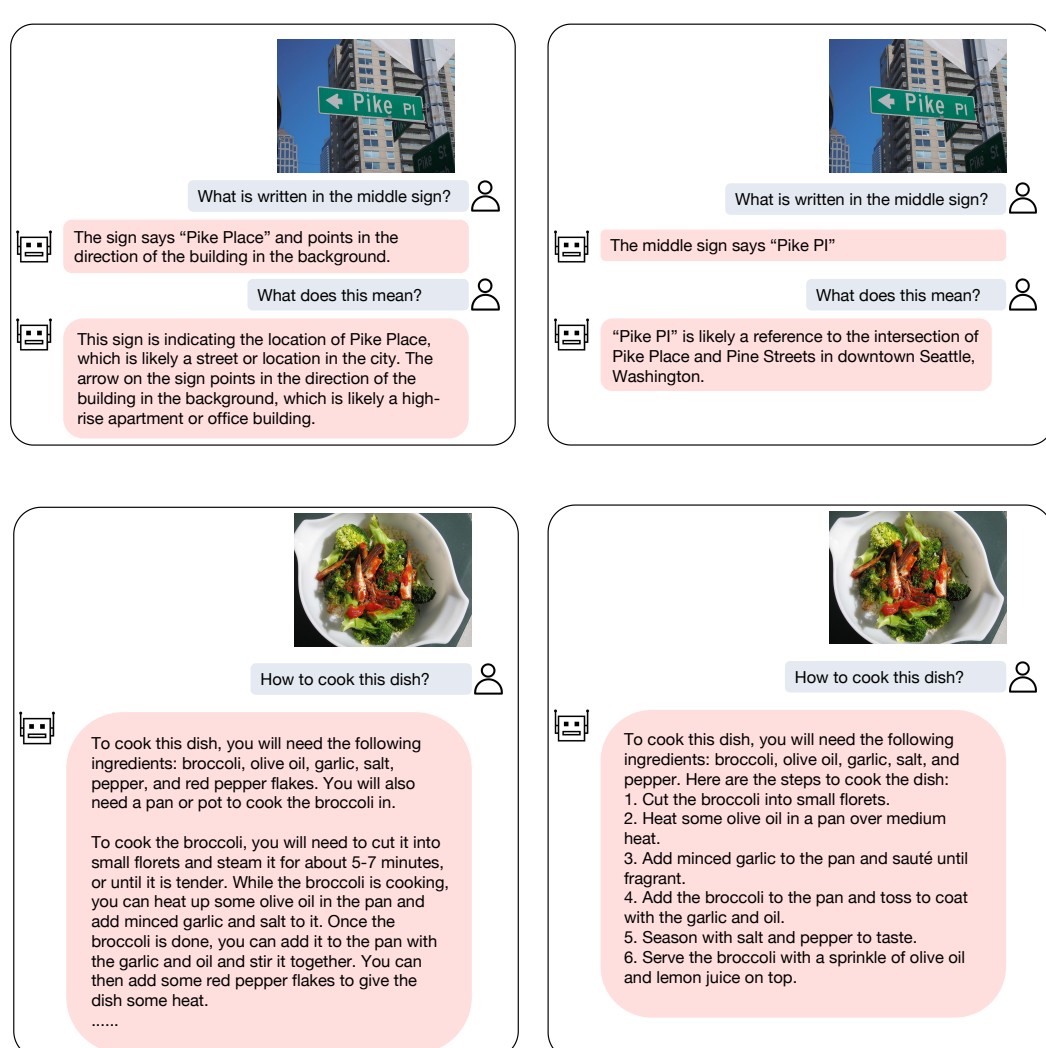

Figure 15: Comparison between MiniGPT-4 and our VL-Vicuna.

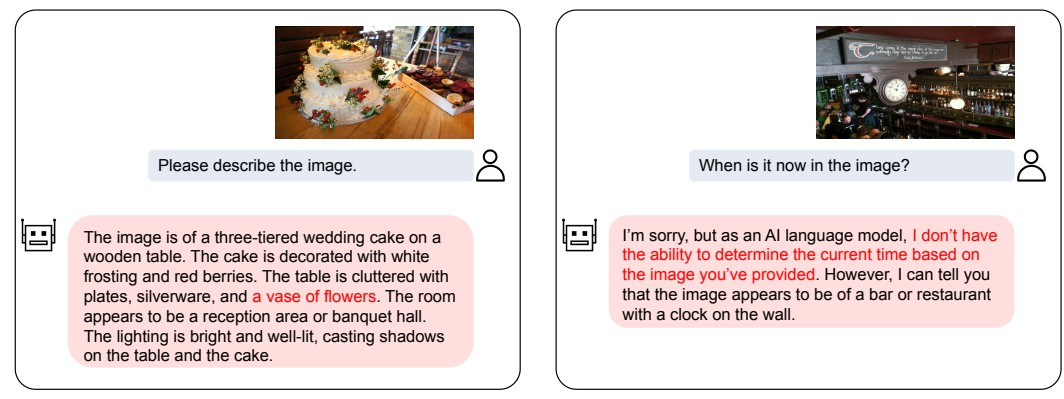

Figure 16: Failure cases of our VL-Vicuna.

