# OpenReview forum: "VPGTrans: Transfer Visual Prompt Generator across LLMs"
_NeurIPS.cc/2023/Conference — NeurIPS 2023 poster_

### Official Review · Reviewer_iPdz · 2023-06-25

**Soundness:** 4 excellent
**Presentation:** 3 good
**Contribution:** 3 good
**Rating:** 6
**Confidence:** 4

**Summary:**

This paper mainly focuses on the transferability of visual prompt generator in VL-LLM. The authors conduct extensive experiments among different LLM types and sizes. By combining the experiment results they propose a new VPN training pipeline that engages projector warmup stage and vanilla finetuning stage.

**Strengths:**

1. This paper is well written.
2. The experiments are comprehensive.


**Weaknesses:**

1. I wonder if different structures for the projector, e.g. more layers, would lead to large performance gap.
2. It is kind of weird to directly compare the training time between training the whole model and using the proposed transfer method, since you have to first get another VL-LLM. In this way the time used for training the original smaller VL-LLM should also be considered. It is inappropriate to claim the method is more efficient just because of other off-the-shelf models.


**Questions:**

Please refer to the above weaknesses.

**Limitations:**

No obvious negative societal impact observed.

---

> ### Author Rebuttal · Authors · 2023-08-08
>
> We sincerely thank you for your time and constructive reviews, which will definitely help consolidate our paper. We are also grateful that you acknowledge the strengths of our work. Following, we present the response to address your concerns.
>
> ---
> **Q1: I wonder if different structures for the projector, e.g. more layers, would lead to large performance gap.**
> **A:** Thanks for your suggestion.
> First of all, we want to clarify that existing VL-LLMs mainly adopt a linear projector and thus we just follow them in this paper.
> Of course, it is quite interesting to explore different projector structures.
> We compare 3 types of projectors (linear, 3 layers MLP, 1 layer transformer) under OPT$_{\text{125M} \rightarrow \text{1.3B}}$ scenario.
> To evaluate the acceleration speed, we compare the performance of COCO caption (CIDEr) at different epochs as follows:
>
> |Projector Type| 1     | 2     | 3     | 4     | 5     | 6     | 7     | 8     | 9     | 10    |
> |-|-------|-------|-------|-------|-------|-------|-------|-------|-------|-------|
> |Linear| 131.5 | 133.1 | 134.1 | 135.1 | 135.7 | 135.9 | 136.1 | 136.2 | 137.0 | 136.8 |
> |3 Layers MLP| 96.8  | 120.9 | 125.5 | 126.7 | 128.3 | 129.4 | 130.4 | 129.7 | 131.5 | 131.9 |
> |Transformer| 132.3 | 134.7 | 134.7 | 134.8 | 135.6 | 135.4 | 135.8 | 135.0 | 135.9 | 135.7 |
>
> To compare the visual perception ability, we show their best VQA performances:
>
> |Projector Type| VQAv2 (acc.) |
> |-|--------------|
> |Linear| 48.9         |
> |3 Layers MLP| 49.6         |
> |Transformer| 49.2         |
>
> We have the following observations:
> - Linear can achieve fast convergence, while the VQAv2 performance is the lowest.
> - 3 layers' MLP will result in much slower convergence and weak COCO caption performance, while achieves the best VQA results after convergence.
> - Transformer can achieve both good convergence speed and good performance on COCO caption and VQAv2,
> which may serve as a potential exploration direction for the projector structure design.
>
> Overall, thanks for indicating this point. We will further explore it in the revision.
>
> ---
> **Q2: It is kind of weird to directly compare the training time between training the whole model and using the proposed transfer method, since you have to first get another VL-LLM. In this way the time used for training the original smaller VL-LLM should also be considered. It is inappropriate to claim the method is more efficient just because of other off-the-shelf models.**
> **A:** Thanks for your careful reading and for mentioning this aspect. Here we report the source models' GPU hours as follows:
>
> | Model                      | Src Model Cost (hours) | Transfer Cost (hours) |
> |-|-|-|
> | BLIP-2 OPT 6.7B            | 631.5                  | 0                     |
> | VPGTrans OPT 6.7B (ours)   | 459.0                  | 59.0                  |
> | BLIP-2 FlanT5 XXL          | 684.0                  | 0                     |
> | VPGTrans FlanT5 XXL (ours) | 435.0                  | 32.4                  |
>
> Compared with training a large model from scratch, our VPGTrans enables training both the smaller and the larger models with even less time.
> We will add the table in the revision.
>
> Moreover, we would like to re-emphasize that the existence of off-the-shelf models is not a demanding assumption.
> When building a new VL-LLM, it is common to validate on a smaller model first, which can serve as a transfer source.
> There are also existing open-sourced VL-LLMs like BLIP-2, which can serve as the base for cross LLM types transfer.

---

> > ### Comment · Reviewer_iPdz · 2023-08-18
> >
> > Thank the authors for the detailed response. I have no further questions.

---

### Official Review · Reviewer_mi8b · 2023-07-02

**Soundness:** 4 excellent
**Presentation:** 3 good
**Contribution:** 4 excellent
**Rating:** 9
**Confidence:** 5

**Summary:**

The paper discusses the transfer of a visual prompt generator (VPG) across different vision-language language models (VL-LLMs) to reduce computational costs. VL-LLMs include a VPG module that bridges the gap between vision and language, encoding visual inputs into fixed-length soft prompts. To reduce the cost of building new VL-LLMs, the authors propose a two-stage transfer learning framework, VPGTrans, for VPG transfer across different LLM sizes and types. VPGTrans includes a projector warm-up (stage-1) and vanilla fine-tuning (stage-2). They conduct exploratory analyses to determine key factors for transfer efficiency and show that VPG transfer across frozen LLMs is feasible and can lead to substantially lower computational costs. The authors demonstrate the practical value of VPGTrans by customizing two novel VL-LLMs with recently released LLMs. The paper highlights the need for further research into VPG transfer across LLMs to further reduce the cost of building new VL-LLMs. Overall, the paper's main contributions include identifying the feasibility and effectiveness of VPG transfer across LLMs, smaller LLM sizes leading to more speed-up and better performance during transfer across LLM sizes, and VPG trained on smaller LLMs outperforming larger ones in most conditions, and then propose a transfer learning framework with empirical evidence supporting its efficiency.

**Strengths:**

1. The multimodal large language model (MM-LLM) is considered the most promising path towards achieving Artificial General Intelligence (AGI). However, a major challenge is the significant computational cost associated with training an MM-LLM. This obstacle hinders the widespread research and exploration of this topic. The presented paper offers an excellent pilot study that demonstrates the feasibility of VPG transfer. By showcasing the viability of VPG transfer, the paper provides valuable insights into mitigating the computational cost of training MM-LLMs. This finding has significant implications for future research in this field. The pilot study serves as a promising starting point, paving the way for further investigations and advancements in the development of MM-LLMs with higher efficiency.

2. I quite enjoy the didactic/exploratory nature of this paper. I like that the paper reads like an investigation, and begins with a preliminary exp, as well as diagnosing reasons for the case, before then presenting the VPGTrans method. And then they delve into the transfer of VPGs under two settings. This, from the shallower to the deeper, feels more organic and the lessons learned along the way are insightful. Also the introduction with clear bullet point allows easy reading.

3. The proposed method, VPGTrans, can be fairly straightforward but effective, which is a merit.

4. The experimental work in this study is extensive and solid. It includes validation across a wide range of LLM sizes and types, providing thorough coverage.

5. This work yields numerous interesting and meaningful findings, which will serve as important empirical evidence for future explorations in LLM/MM-LLM efficiency. The conclusions drawn from this research have the potential to guide and shape subsequent investigations in this field.

6. Additionally, this paper makes good contributions by introducing two new state-of-the-art MM-LLMs in the community: VL-LLaMA and VL-Vicuna. Overall, I like this paper, and I believe this work will show important impact to the community.

**Weaknesses:**

1. The major possible limitation of this method is its reliance on the existence of a well-functioned VPG model, and without this assumption, this work is of less meaningfulness. How does the quality of the existing VPG influence the transfer efficacy and efficiency?

2. This work evaluates on merely the captioning and VQA tasks. It would be interesting to see the performance on other different VL tasks and more benchmark datasets, to see the broad-coverage trends.

3. On the other hand, the proposed VPGTrans, though straightforwardly simple, relies too much on the empirical tuning, e.g., when to inherit the raw VPG and when to tuning or fixing some modules. Are there any intuitions and theoretical supports to build up the method?

4. When transferring the VPG, what has been changed of the visual soft prompts? Are the visual prompts of the raw VPG changed, to adapt to the new LLM backbone?

5. The claim of ‘our VPGTrans helps achieve a BLIP-2 ViT-G OPT2.7B→6.7B transfer with less than 10% of the GPU hours and 10.7% training data required for the original model training’ can be a little misleading, as the comparison between VPGTrans and the training of the raw VL-LLM is not strict fair and counterpart; should compare the VPGTrans with the hard VPG transfer method and then draw the conclusion.

6. Lack of exploration of potential limitations or drawbacks of the proposed method, which may limit its generalizability or practical usefulness.

7. How did you choose the specific VL-LLMs for your experiments?

8. In Section 4.2, you mention that the VPG trained on smaller LLMs may outperform larger ones in most conditions. Can you provide more explanation and insights for this observation of why it brought better task performance?

9. Can you explain in more detail the warm-up training of the linear projector and how it can prevent performance drops?

10. In Section 6, you mention that you customized two novel VL-LLMs with your VPGTransfer framework. How generalizable do you think this framework is to other novel VL-LLMs, and what steps would need to be taken to adapt it to different models?

**Questions:**

See Weaknesses.

**Limitations:**

I do not foresee any potential for negative societal impact from this work.

---

> ### Author Rebuttal · Authors · 2023-08-08
>
> We are grateful that you acknowledge the strengths of our work so much. Your support definitely encourages us to improve the work further and push forward. Following we present the response to address your concerns.
>
> ---
> **Q1: Reliance on the existence of a well-functioned VPG model.**
> **A:** Thanks for pointing this out.
> We want to clarify that the existence of VPG model is not a demanding requirement:
> - When building new MM-LLMs, it is common to first validate on small models and then scale up, where the small VPG models can be used for transfer.
> - There are already released VPG models like BLIP-2, which can be used.
>
> ---
> **Q2: How does the quality of the existing VPG influence the transfer efficacy and efficiency?**
> **A:** We discuss the question under TaS and TaT scenarios:
> - TaS: A better VPG can typically result in higher efficacy and efficiency under TaS.
> As shown in Figure 7 of the main paper, the VPG trained on OPT 125M is the best performed one,
> which also achieves the highest acceleration rate in Table 3 and the best performances given the same target LLM.
> - TaT: When considering TaT, there will be a trade-off between the VPG ability and model gap.
> For example, although the VPG trained on smaller OPT shows better results, it is also less transferable to FlanT5 models (cf Appendix D.1).
>
> ---
> **Q3: More tasks and datasets.**
> **A:** Thanks for the suggestion!
> Please refer to the Reviewer#7KJM Q2 and Q3 for results on more datasets (medical, robustness VQA).
>
> ---
> **Q4: Intuition of VPGTrans.**
> **A:** We first dissect the success of VPGTrans into 3 parts and explain the intuitions one by one:
> - 1. Projector warm-up can accelerate normal pre-training (stage-2) and prevent performance drop.
> Intuitions: (1) projector warm-up can work as a good weight initialization for the stage-2 and thus accelerate. (2) the gradient passed through a randomly initialized projector will distort VPG's ability (performance drop), while the warm-up can avoid.
> - 2. Word converter initialization can accelerate projector warm-up. Intuition: a good initialization can lead to faster convergence.
> - 3. Projector enables fast convergence with extremely large learning rate. Intuition: projector has very limited parameters, and thus easier to train.
>
> ---
> **Q5: Theoretical supports of VPGTrans.**
> **A:** In this paper, we mainly focus on the empirical study. We try to discuss some potential theoretical analysis directions that may help:
> - Warm-up Aspect: [1] illustrates that training the classifier first and then fine-tuning can help the OOD performance.
> The paper shows that gradient passed through a randomly initialized classifier will distort pretrained features.
> However, [1]'s theoretical analysis is limited to linear feature extractor assumption.
> - Visual Prompt Aspect: there are limited theoretical works on the mechanism of visual soft prompt.
> For empirical study, [2] and our work try to interpret visual prompts with word embeddings (cf Appendix Fig. 11).
>
> ---
> **Q6: When transferring the VPG, what has been changed of the visual soft prompts?**
> **A:** Good question! We visualize the visual prompts (cf Appendix Fig. 11), where the last token is always nearest to EOS token of OPT,
> and other tokens represent visual contents like flower.
> After transferring to larger OPT (e.g. OPT 2.7B), the last token is also nearest to EOS token.
> However, the nearest neighbour of each content token for a given image will be changed to other content (e.g. flower->table).
>
> **Q7: Comparison with hard VPG transfer.**
> **A:** We show the result of hard VPG transfer with similar amounts of training data:
>
> | Model (OPT 6.7B)  | VQAv2 | GQA  | OKVQA | Data  |
> |-------------------|-------|------|-------|-------|
> | Hard VPG Transfer | 50.9  | 33.9 | 32.7  | 13.8M |
> | VPGTrans (ours)   | 57.2  | 36.2 | 39.8  | 13.8M |
>
> We see that VPGTrans can achieve better performance with the same amount of training data.
> We will also add it in the revision.
>
> ---
> **Q8: Limitations and Drawbacks**
> **A:** We discuss the social impact and limitations in the lines 584-594 of Appendix.
> The limitation mainly includes the hallucination problem of constructed VL-LLMs.
>
> ---
> **Q9: How did you choose the specific VL-LLMs for your experiments?**
> **A:** For the VL-LLM architecture, we choose BLIP-2, as it is the most powerful open-sourced MM-LLM when we initialize this project.
> For the LLM choices, we try to cover both encoder-decoder (FlanT5) and decoder-only (OPT) models across different sizes.
>
> ---
> **Q10: VPG trained on smaller LLMs may outperform larger ones. Potential explanations.**
> **A:** First of all, training VPG on LLMs will change the weight of the VPG, which will affect the pretrained VPG's ability. This phenomenon can be considered as a kind of VL alignment tax.
> Then, we empirically find that using larger OPT model usually takes more tax.
> The conclusion can be validated by the confusion matrix (cf Figure 7), where the VPG is fixed to see which VPG is better.
> We also check the absolute update of transfer sources (OPT 125M, 350M, 1.3B) and find that the VPG of OPT-1.3B has the largest absolute update in the 1st epoch.
> Finally, it is still unclear why the larger LLM will have more effect on VPG.
> We hypothesize it might be caused by the more complicated mechanism, which may require more effort for VPG to align.
> We will further explore this problem in the future.
>
> ---
> **Q11: Why warm-up can prevent performance drop.**
> **A:** Please refer to Q4-1-(2).
>
> ---
> **Q12: Adaptions to different models.**
> **A:** The adaptation will just be similar to the construction of VL-LLaMA and VL-Vicuna.
> - Find an existing VPG (e.g. ImageBind-LLM's VPG), and an LLM (e.g. LLaMA-2).
> - Then connecting them with VPGTrans' two-stage training.
>
> ---
> **Q13: Potential Social Impacts.**
> **A:** Please refer to Q8.
>
> ---
> [1] Fine-Tuning can Distort Pretrained Features and Underperform Out-of-Distribution.
> [2] CLIPCap: CLIP Prefix for Image Captioning.

---

> > ### Comment · Reviewer_mi8b · 2023-08-12
> > **Response to author rebuttal**
> >
> > Thank you for addressing my inquiries. The majority of my concerns have been satisfactorily resolved, and I generally concur with the authors' assertions.
> >
> > Furthermore, regarding one of the points posed by Reviewer #GwBd, I would like to inquire whether the overall training pipeline might become intricate and challenging to implement in practical scenarios. I've observed that the proposed framework involves a substantial number of pipelines, have these been automated when using it? Do I need to care about the intermediate operations?
> >
> > In regard to one of my questions (Weakness#5), specifically the comparison between VPGTrans and hard VPG transfer, I would greatly appreciate a comprehensive analysis encompassing factors about operational efficiency, for instance, execution time and costs. Currently, you have presented the end task performance instead of the efficiency comparisons.

---

> > > ### Author Response · Authors · 2023-08-13
> > > **Re-rebuttal to Reviewer #mi8b**
> > >
> > > Thanks for your reply!
> > > It is encouraged to see that you are satisfied with our rebuttal.
> > > In the following, we try to address your current concerns point to point.
> > >
> > > ---
> > > **Q1: Whether the overall training pipeline might become intricate and challenging to implement in practical scenarios.
> > > I've observed that the proposed framework involves a substantial number of pipelines, have these been automated when using it?
> > > Do I need to care about the intermediate operations?**
> > >
> > > **A:** Thanks for the question.
> > > Actually, our VPGTrans is quite simple to implement.
> > > - In the code level, the main workload lies in the word converter trainer implementation,
> > > while the code can be implemented by adding a new file without modifying the original VL-LLM code.
> > > Thus, the practitioners do not need to dive into the implementation details of the original VL-LLM to modify it.
> > >
> > > - In the experiment level, the framework can be automated without considering the intermediate results.
> > > Some researchers may concern about the hyperparameters about the intermediate operations.
> > > We want to clarify that the main hyperparameter, the 1st stage's learning rate, is super robust
> > > in the training (cf lines 524-528 Appendix), which does not require careful adjustment.
> > >
> > > ---
> > > **Q2: I would greatly appreciate a comprehensive analysis encompassing factors about operational efficiency, for instance, execution time and costs. Currently, you have presented the end task performance instead of the efficiency comparisons.**
> > >
> > > **A:** Sorry for missing your point.
> > > We have a comparison between VPGTrans and VPG transfer (denoted as VPG inherit in our paper) in **C.3 of our Appendix**,
> > > where the speed-up rate is reported in Table 5 and the convergence curve comparison is shown in Figure 13.
> > > A main conclusion is that our VPGTrans can achieve better performance for over 74% Transfer-Task variants,
> > > among which **our VPGTrans can also achieve better acceleration rate for over 69% conditions**.
> > > More analysis will be added in the revision.

---

> > > > ### Comment · Reviewer_mi8b · 2023-08-14
> > > > **Response to author rebuttal-maintain my initial rating.**
> > > >
> > > > Thank the authors for the response.
> > > > Overall, I think this paper has great research significance and I stick to my previous rating.

---

### Official Review · Reviewer_GwBd · 2023-07-06

**Soundness:** 3 good
**Presentation:** 2 fair
**Contribution:** 3 good
**Rating:** 4
**Confidence:** 5

**Summary:**

* This paper presents an interesting study on how to effectively transfer the small VL model to the large VL model, which is very practical under the fact that the LLMs are very expensive to finetune.
* The empirical study shows the effectiveness of the methods.

**Strengths:**

* The paper proposes an effective word converter to better speed up the model training on a large model
* The stage-wise training gives good results for the final performance.

**Weaknesses:**

* The paper writing is very complex to understand
* This technical contribution is very limited. The core is word converter, which better aligns the knowledge between large LLMs and small LLMs.
* The performance improvement is very obvious. Because the whole model training involves the small LLM, which contains more model parameters.
* The whole training pipeline is very complex. There are multiple stage for training.

**Questions:**

* It is better to include the GPU hours and training data of a small model for a fair comparison
* Is it possible to inject the Alignment loss during the tuning projector stage?


**Limitations:**

see above

---

> ### Author Rebuttal · Authors · 2023-08-06
>
> We sincerely thank you for the valuable suggestions. Following, we present the point-to-point response to address your concerns. And if you feel our responses are effective, please kindly raise your evaluation.
>
> ---
> **Q1: The paper writing is very complex to understand.**
> **A:** Thanks for your comment. We'd like to make a clarification on our paper organization.
> Different from the traditional _introduction-method-experiment_ structure, we adopt an **exploratory structure**.
> The exploratory structure aims at deriving the method from a series of exploratory experiments over existing materials, which should be a merit for the topic of our focus.
> As admired by the Review#mi8b (Strengths 2), _'the paper reads like an investigation, from the shallower to the deeper, feels more organic and the lessons learned along the way are insightful'_.
>
> To make it easier to follow for you, we list the outlines of the paper:
> - Sec. 1: Motivations and paper summary;
> - Sec. 2: Descriptions/preliminaries of VL-LLMs;
> - Sec. 3: Exploratory analysis $\to$ Our proposed method;
> - Sec. 4: Experiments for cross LLM sizes transfer;
> - Sec. 5: Experiments for cross LLM types transfer;
> - Sec. 6: Building new VL-LLMs with our method.
>
> We will further revise the section titles to make each part clearer.
>
> ---
> **Q2: This technical contribution is very limited. The core is word converter, which better aligns the knowledge between large LLMs and small LLMs.**
> **A:**  We kindly re-emphasize our major contributions as to offer a thorough investigation to demonstrate the feasibility of VPG transfer for efficient accessibility of VL-LLMs, and devise a novel transfer method VPGTrans. In summary, our contributions are multifaceted:
>
> - We propose a VPGTrans framework, which significantly reduces the cost of building new VL-LLMs to lab-level resources.
> - We reveal intriguing insights and findings of the VPG transfer, and shed light on further research on this topic.
> - With VPGTrasn, we customize two novel VL-LLMs that are open-sourced for the community.
>
> That being said, although not presenting ground-breaking technical novelty, we do contribute with innovations from the technical perspective of our proposed VPGTrans framework:
> - We proposed the _projector warm-up_ strategy, which helps achieve transfer with both high efficiency and high efficacy.
> - We devised a novel _word converter_ method that can accelerate projector warm-up for over 1.5 times.
> - We also proposed _training the projector with an extremely large learning rate_, via which we find can further reduce the warm-up consumption to 1/3.
>
> Most importantly, we validate the feasibility of VPG transfer with systematical analysis via our VPGTrans system, where the findings help future researchers to obtain better practices in VPG transfer for building VL-LLMs.
>
> We sincerely hope you can re-evaluate our contributions and the significance of this work.
>
> ---
> **Q3: The performance improvement is very obvious. Because the whole model training involves the small LLM, which contains more model parameters.**
> **A:** We respectfully disagree. The truth is, the **small and large models are trained and tested separately** in our scenario; but the scaling advantage you indicated in the comment mainly happens when conducting training and testing with one joint model.
> In fact, it is non-trivial to conduct the transfer with high efficiency and efficacy. For example, the direct transfer (_VPG inherit_) even achieves worse performance than training from scratch.
>
> But we think this is a wise question, and thanks for pointing this out. We will strengthen this point in the revision.
>
> ---
> **Q4: The whole training pipeline is very complex. There are multiple stages for training.**
> **A:** Our method actually contains two stages, and we don't think it is complex if we compare it with current VL-LLMs.
> Recent works on VL-LLMs like LLaVA (2 stages) and mPlug-Owl (2 stages) all entail 2 stages of training.
> In addition, our source codes are made very easy and convenient for practitioners to implement.
> Thus, there will be only very small cost to use our system and apply it into realistic applications.
>
> ---
> **Q5: It is better to include the GPU hours and training data of a small model for a fair comparison.**
> **A:** Thanks for the suggestion. The training costs are shown as follows:
> | Model                      | Src Model Cost (hours) | Transfer Cost (hours) |
> |:-|:-:|:-:|
> | BLIP-2 OPT 6.7B            | 631.5                  | 0                     |
> | VPGTrans OPT 6.7B (ours)   | 459.0                  | 59.0                  |
> | BLIP-2 FlanT5 XXL          | 684.0                  | 0                     |
> | VPGTrans FlanT5 XXL (ours) | 435.0                  | 32.4                  |
>
> The total training data of the small model + transfer is the same as the larger ones.
> Compared with the one training from scratch, our VPGTrans can use even less time to train both the smaller and larger models.
> We will add the table in the revision.
>
> ---
> **Q6: Is it possible to inject the Alignment loss during the tuning projector stage?**
> **A:** Thanks for the question, good idea!
> We try to discuss different ways for the alignment loss.
> We list the elements when conducting the transfer: (1) source VPG (2) source LLM (3) target LLM, where the alignment loss can be designed as these:
> - source VPG$\to$source LLM: already aligned.
> - source VPG$\to$target LLM: it is worth exploring the **alignment between visual soft prompts and word embeddings**.
> Actually, current visual soft prompts (visual features) generated by VPG are not fully aligned with the word embedding.
> While the cosine similarity is high sometimes, the norm of two types of embeddings are quite different (Appendix line 519-520).
> Thus, it will be interesting to explore the influence of alignment with different distance metrics.
> - source LLM$\to$target LLM: it is what the word converter does.
> We will discuss this in the revision.

---

> > ### Author Response · Authors · 2023-08-16
> > **We are looking forward to your feedback**
> >
> > Dear Reviewer#GwBd,
> >
> > Thanks so much for your great efforts and valuable feedback on our paper.
> > Your comments are essential to help us improve the quality of our work.
> > We have carefully addressed your concerns and questions in our responses.
> > For example, we report the cost of training the source model for a fair comparison,
> > and we systematically discuss the possibility of incorporating the alignment loss.
> > We kindly hope that you can take some time to re-evaluate our paper based on our replies.
> > If you have any further concerns or questions, please do not hesitate to let us know.
> > We will be happy to address them promptly.
> >
> > Best Regards,
> > Paper#1904 Authors

---

### Official Review · Reviewer_7KJM · 2023-07-11

**Soundness:** 2 fair
**Presentation:** 3 good
**Contribution:** 2 fair
**Rating:** 6
**Confidence:** 4

**Summary:**

The paper presents a technique for VPG (Visual Prompt Generator) transferability across LLMs where the transferability can be between LLMs of different sizes (eg: OPT125M -> OPT2.7B) or across LLMs of different types (eg: OPT350M-> Flan T5base). This is achieved using a two stage strategy where a projector is learned by freezing VPG and LLM in the first stage and vanilla finetuning of VPG and projector in the second stage. The proposed technique significantly reduce training time (specifically when transfer from small to large LLMs).

**Strengths:**

1. The VPG transferability can significantly improve the training time/cost in some cases (from LLM_small->LLM_large)  can have many practical applications when training larger LLMs.
2. The proposed technique consistently reduces the training time/cost compared to baseline (TFS) while achieving similar performance

**Weaknesses:**

Novelty: The proposed technique is a standard technique used in fine-tuning a model for a specific task where the newly added layer (head which is similar to projector here) is initially trained with backbone frozen and once trained, the whole network (backbone and head) is trained/fine-tuned on the target data. Please see [a, b] for references that talk about standard fine-tuning strategies. The authors extend this to VPG transferability between two LLMs. Although, the technique is good for practitioners, I don't see any technical insights in the proposed technique.

Generalization: Since the proposed technique uses a limited vocabulary from the datasets COCO, SBU-captions [1.4M image-caption pairs] for the word converter, the performance of the target LLM on the datasets with domain GAP could be affected. It wold be interesting to see the performance of the target LLM (and comparison with source LLM) on domain specific datasets such as RSVQA [c] and PathVQA [d] datasets.

Robustness: To check the validity of the proposed technique, It would also be interesting to see the affect of transferability on Robustness of the target model which [target-LLM compared src-LLM] can be done by evaluating on datasets AdVQA [e], VQA-CE [f] or VQA-Rephrasing [g]


[a] https://cv-tricks.com/keras/fine-tuning-tensorflow/

[b] https://lightning-flash.readthedocs.io/en/stable/general/finetuning.html

[c] https://rsvqa.sylvainlobry.com/#overview

[d] https://paperswithcode.com/dataset/pathvqa

[e] https://adversarialvqa.org/

[f] https://paperswithcode.com/dataset/vqa-ce

[g] https://facebookresearch.github.io/VQA-Rephrasings/

**Questions:**

I would like authors to discuss the three points raised above : 1) Novelty, 2) Generalization, [3] Robustness

---

> ### Author Rebuttal · Authors · 2023-08-06
>
> We sincerely thank you for your time and the careful review. Your suggestions will definitely help improve our paper. Following, we present the response to address your concerns. And if you feel our responses effectively relieve your concerns, please kindly reconsider your evaluation.
>
> ---
> **Q1: Novelty: The proposed technique is a standard technique used in fine-tuning... Although, the technique is good for practitioners, I don't see any technical insights in the proposed technique.**
> **A:** While it is a fact that the transferring technique in our work coincides with the existing standard method for the practices of transfer learning, we would like to argue that our work is far more than the simply leverage of the model transfer itself.
> - First, we have actually made certain technical improvements (simple yet effective) for the scenario of LLM transfer over the existing 'standard transfer learning'. For example, we devise the **word converter based initialization** (cf Sec. 3.2), without which, the warm-up of the projector will be 1.5 times slower. On the other hand, getting rid of complicated architectural design, we rather believe being simple to use to achieve prominent efficiency improvement is a merit of our method.
> - Second, compared to the technical innovation, our key contribution/novelty more lies in the experimental explorations. We first performed systematical and in-depth analyses to confirm the feasibility, before we decided to conduct the VPG transfer and propose our method. We note that, without such preliminary explorations, the direct transfer of VPG will be significantly less effective compared to ours.
> - Third, we for the first time presented a rich amount of intriguing findings/insights and meaningful explanations for the scenario of VPG transfer across LLMs, which will largely pave the way for all the furture explorations in this topic. For example, we show that, for OPT based VL-LLMs, the VPG trained on smaller src-LLM typically yield better transfer result.
> - Finally, our contributions to the community are substantial and evident, especially for the practical applications in production environment. Our VPGTrans enables building new VL-LLMs with much reduced cost (e.g. 10% cost), making it possible to **build new VL-LLMs with lab-level resources**. The VL-LLaMA and VL-Vicuna presented in our paper are good examples. For example, for some startup companies want to build their own VL-LLMs, our VPGTrans offers a valuable path to **save thousands and millions of dollars**.
>
> ---
> **Q2: Generalization: Since the proposed technique uses a limited vocabulary from the datasets COCO, SBU-captions for the word converter, the performance of the target LLM on the datasets with domain GAP could be affected. It would be interesting to see the performance of the target LLM (and comparison with source LLM) on domain-specific datasets such as RSVQA and PathVQA datasets.**
> **A:**  Thank you for carefully going through our paper. We want to clarify that our model is also trained on millions of web images (similar to BLIP-2's data composition) during **VPGTrans stage-2** (after the word converter training), i.e., the model sees data in multiple domains, which ensures a good generalizability over domains.
> Moreover, the fact is that boosting the amount of training data of word converter can be quite cheap and effortless. For example, using 100M data for 1 epoch training will only take **<40 min** consumption with one A100 (80G).
>
> During the rebuttal days, we run the experiments on domain-specific datasets for the LLMs. The experiments are with FlanT5-XL (src) and FlanT5-XXL (tgt):
>
> | Model | Data   | PathVQA  (yes&no acc. / F1) | RSVQA (acc.) |
> | :-------:|:--------:|:-----------------------------:|:--------------:|
> | src   | 121.6M | 50.6/27.2                   | 40.3         |
> | tgt   | 5.3M   | 51.2/26.9                   | 41.7         |
> | tgt   | 9.4M   | 51.3/27.0                   | 44.0         |
>
> And we find that our model can achieve even higher/comparable performance on these two datasets, compared with the source model.
> Moreover, we notice that including more data can further improve the VQA results on other domains.
> Thanks for indicating this, and we will consider adding it into revision.
>
> ---
> **Q3: Robustness: To check the validity of the proposed technique, It would also be interesting to see the effect of transferability on Robustness of the target model which [target-LLM compared src-LLM] can be done by evaluating on datasets AdVQA, VQA-CE or VQA-Rephrasing.**
> **A:** Thanks for the suggestion, good idea.
> Also, we run the experiment on the datasets you indicated, and the results are as follows. Likewise, the experiments are with FlanT5-XL (src) and FlanT5-XXL (tgt):
>
> | Model | Data   | AdVQA (acc.) | VQA-CE (acc.) | VQA-Reph. (acc.) |
> | :-------:|:--------:|:-----------------------------:|:--------------:|:--------------:|
> | src   | 121.6M | 37.2         | 36.7          | 60.2             |
> | tgt   | 5.3M   | 39.7         | 39.0          | 62.8             |
> | tgt   | 9.4M   | 40.3         | 39.4          | 63.2             |
>
> Compared with the source model, our tgt model can achieve better performance on all of three VQA datasets, demonstrating its capability on robustness.
> This part will be added into revision, thanks again.

---

> > ### Author Response · Authors · 2023-08-16
> > **We are looking forward to your feedback**
> >
> > Dear Reviewer#7KJM,
> >
> > We would like to express our sincere appreciation for your great efforts and valuable feedback on our paper.
> > Your comments are essential to help us improve the quality of our work.
> > To address your concerns of generalization and robustness, we compare the source model and target model on RSVQA, PathVQA, AdVQA, VQA-CE, and VQA-Rephrasing datasets.
> > We kindly hope that you can take some time to re-evaluate our paper based on our replies.
> > If you have any further concerns or questions, please do not hesitate to let us know.
> > We will be happy to address them promptly.
> >
> > Best Regards,
> > Paper#1904 Authors

---

> > > ### Comment · Reviewer_7KJM · 2023-08-21
> > > **Final**
> > >
> > > I have read the rebuttal and also other comments. I am happy with the experiments on Robustness and generalization. So I am increasing my score.

---

### Author Rebuttal · Authors · 2023-08-06

# General Response to All Reviewers
Dear reviewers,

Thanks for all of your time to write valuable and constructive comments. Your feedback will definitely assist us in enhancing the quality of our paper, and thus we are committed to incorporating your suggestion in our revision process.
Meanwhile, we feel encouraged that the reviewers find our method efficient and effective (Reviewer 7KJM, GwBd and mi8b), and our experiments solid and comprehensive (Reviewer iPdz and mi8b). Your support means a lot to us!
At this juncture, we would like to re-emphasize the significance of this work.

The AI community has now entered the era of Large Language Models (LLMs), wherein multimodal Vision-Language LLMs (VL-LLMs) have demonstrated a powerful understanding across vision&text modalities, becoming a focal point of future LLM research. However, we recognize that obtaining a VL-LLM can be costly, especially when training from scratch. This has motivated our work: **exploring the model transfer learning approaches to significantly reduce the cost of acquiring VL-LLMs**, such as transitioning from existing smaller models to larger models.
With such background, this work contributes to the following key aspects:

1. We propose a VPGTrans framework, which significantly reduces the cost of building new VL-LLMs (e.g. in only 10% GPU hours) with the help of existing bases.
2. We reveal intriguing insights and findings of the VPG transfer across LLMs, and provide potential explanations to shed light on further research on this topic.
3. With VPGTrasn, we customize two novel VL-LLMs (i.e., VL-LLaMA and VL-Vicuna) that are open-sourced for the community.

Although this paper may not present ground-breaking technical novelty (pointed out by Reviewer#7KJM), we mainly contribute by offering a thorough investigation to demonstrate the feasibility of VPG transfer for efficient accessibility of VL-LLMs, and devise a novel transfer method **VPGTrans**.
As recognized by Reviewer#mi8b, _'the pilot study serves as a promising starting point, paving the way for further investigations and advancements in the development of MM-LLMs with higher efficiency'_.
Our VPGTrans enables more swift building of new VL-LLMs with lab-level resources, and thus will enable more researchers into this area.
We much believe VPGTrans will show a broad impact on the future research of LLMs. Thus, we will release all the codes and resources upon acceptance.


In response to the reviewers' comments, we have thoroughly reviewed our paper, performed additional experiments, and prepared a comprehensive response.
We will fix all the typos and improve the manuscript according to your comments.
We hope that our paper adequately addresses your concerns.
We kindly request a **re-evaluation of our work** based on the updated information, and look forward to your recognition.

Best regards.

---

### Comment · Area_Chair_1uWB · 2023-08-18

Hi Authors,

Thanks a lot for your rebuttal. We are urging all the reviewers to respond and will take your input into consideration as we make the final recommendation. Thanks!

Best,

---

### Decision · Program_Chairs · 2023-09-21

**Decision:**

Accept (poster)

**Comment:**

The paper received 6/9/6/4 ratings. Reviewers acknowledge that the experimental findings in VPG (Visual Prompt Generator) transfer from one VL-LLM to another VL-LLM may benefit the community. The score 4 review raised concerns regarding the presentation of the paper as well as limited technical contribution. The AC read the paper, the reviews as well as rebuttal discussions, and agreed with part of the concerns. Although it’s straightforward that initializing the VPG with a well-trained existing VPG can significantly reduce the training time when adopting to a new LLM (limited technical contribution), the study could be viewed as providing a best practice for this. Thus the AC recommends to accept the paper.